# Risk communication during seismo-volcanic crises: the example of Mayotte, France

Maud Devès[1,2], Robin Lacassin[1], Hugues Pécout[3], Geoffrey Robert[1]

[1] Université de Paris, Institut de physique du globe de Paris, CNRS UMR 7154, 75005 Paris, France, contact author: deves@ipgp.fr

[2] Université de Paris, Institut Humanités Sciences Sociétés, Centre de Recherche Psychanalyse Médecine et Société, CNRS EA 3522, 75013 Paris, France

[3] Université de Paris, Collège international des sciences territoriales, CNRS FR 2007, 75013 Paris, France

# Abstract

Population information is a fundamental issue for effective disaster risk reduction. As demonstrated by numerous past and present crises, implementing an effective communication strategy is however not a trivial matter. This paper draws lessons from the seismo-volcanic "crisis" that began in the French overseas department of Mayotte in May 2018 and is still ongoing today.

Mayotte's case study is interesting because: i) although the seismo-volcanic phenomenon itself is associated with moderate impacts, it triggered a social crisis that risk managers themselves qualified as "a communication crisis", ii) risks are perceived mostly indirectly by the population, which poses specific challenges, in particular to scientists who are placed at the heart of the risk communication process, iii) no emergency planning or monitoring had ever been done in the department of Mayotte with respect to volcanic issues before May 2018, which means that the framing of monitoring and risk management, as well as the strategies adopted to share information with the public, have evolved over time.

Our first contribution is to document the gradual organisation of the official response. Our second contribution is an attempt to understand what may have led to the reported "communication crisis". To that end, we collect and analyse the written information delivered by the main actors of monitoring and risk management to the public over the last three years. Finally, we compare its volume, timing and content with what is known of at-risk populations information needs. Our results outline the importance of ensuring that communication is not overly technical, that it aims to inform rather than reassure, that it focuses on risk and not only on hazard and that it provides clues to possible risk scenarios. We finally issue recommendations for improvement of public information about risks, in the future, in Mayotte, but also elsewhere in contexts where comparable geo-crises may happen.

# 1.    Introduction

As recalled by the Sendai Framework for Disaster Risk Reduction, population information is a fundamental issue for effective disaster risk reduction (UNISDR, 2015, article 18.g.). Some researchers even consider that a disaster is a result of a crisis or a breakdown in the communication process (e.g. Gilbert, 1998). Implementing an effective communication strategy is however not a trivial matter. As pointed out by previous studies, and as exemplified by the current COVID-19 crisis, there are numerous pitfalls (see Lagadec, 1993; Lindell, Prater and Perry 2006 or Rodriguez et al., 2007 for overviews). Deciding what content, format and medium to use to share information is a first challenge. The information held by the actors in charge of risk management is often partial, sometimes contradictory, especially at the beginning of a crisis when there are many unknowns; the information available - and especially the information produced by scientists - can be difficult to translate into operational terms when there are large uncertainties; actors might also have difficulties in sharing information and/or in coordinating (see Doyle and Paton, 2018; Donovan, Bravo and Oppenheimer, 2012; Donovan and Oppenheimer, 2012; Fearnley and Beaven, 2018 for application on volcanic risks). Reaching the population at-risk is a next challenge. Traditional channels (press releases, public conferences, mass media) may allow reaching a majority of people, but might not help reaching minorities whose habits, customs, and sometimes day-to-day language, differ (Lindell and Perry, 2004). And, it is not enough for a message to reach people, it must then be understood, believed and confirmed to have a chance to induce the expected response (e.g. Mileti and Sorensen, 1990; Mileti, 1999; Lindell and Perry, 2004). This implicitly raises the issue of trust and of the perceived credibility and legitimacy of information providers (see Haynes, Barclay and Pidgeon, 2008 for a reflection on the importance of trust in the management of volcanic risks).

The present paper contributes to the effort made by human and social sciences to build knowledge on risk communication processes. It draws lessons from the seismo-volcanic "crisis" that began in the French overseas department of Mayotte in May 2018 and is still ongoing at the time of writing. It focuses on "public information" i.e. on the information shared by the actors in charge of monitoring and risk management with the public. The corresponding processes are sometimes called "external" communication processes, "internal" communication referring to the exchanges taking place between the actors (e.g. Becker et al., 2018).

Mayotte's case study is interesting because, although the seismo-volcanic phenomenon itself has been associated with moderate impacts (see section 2), it triggered a social crisis that the risk managers themselves qualified as "a communication crisis" (see section 3). The situation has eased in part nowadays but scientists and authorities are still regularly taken to task, especially on social media (see section 5). Mayotte's case study is also interesting because, with the exception of felt seismicity, deep sea dead fishes occasionally found by fishermen, and gas bubbling in a few spots on land, risks are perceived mostly indirectly by the population at risk. As Skotnes, Hansen and Krovel (2021) point out, risk and crisis communication about "invisible" hazards poses specific challenges. While trust is a key factor in communication in general, it becomes all the more crucial when one must rely entirely on the knowledge and experience of others to make decisions. The seismo-volcanic phenomena at stake here are not, strictly speaking, "invisible" (not in the sense of chemical or radiological pollution for instance) but

everything one knows about it comes from scientific observation and interpretation. This puts
scientists at the heart of the risk communication processes. Public information emerges thus in
Mayotte, more than ever, as an end product of a complex interface between science, policy and
society. Decrypting this interface's mechanisms and dynamics is necessary to help actors,
including scientists, better understand their role and its limits[1].

Scientists and authorities have complementary roles to play with respect to population
information. The local and national authorities are in charge of informing populations at risk about
the nature and evolution of the threat and about the measures put in place to manage or reduce
it. Scientists have a key role to play in helping the other stakeholders of the "risk chain", including
the at-risk population and the wider public, to comprehend scientific information as the latter is
often too technical for non-specialists (e.g. Newhall, 1999; Fearnley and Beaven, 2018). This role
is essential to maintain the legitimacy and credibility of the information on which public decisions
are based, scientists being generally more trusted than their official counterparts (e.g. Eiser et al.,
2008 on the predictors of trust and Donovan, 2021 for an overview of the challenges faced by
experts in crisis contexts).
In Mayotte, as far as seismo-volcanic risk is concerned, a disaster has not yet occurred -
the seismic crisis, although worrying for the population, has not caused significant damage. But
many questions remain unanswered concerning the potential effects of the current activity in the
short or medium term (see section 2). Today's challenges are therefore those of scientific
research to understand, monitoring to alert, and prevention and preparedness to reduce potential
impacts, improve emergency management, and foster individual and collective resilience. As a
recent report commissioned by the French ministry in charge of risk management (*ministère de*
*la Transition écologique et solidaire*) reminds us, the involvement of the population is crucial for
the success of the process as a whole (Courant et al., 2021). There are, however, several
indications that Mayotte's inhabitants have not been satisfied with the way information has been
shared about the current event (see section 3). Although, as we will demonstrate later on, there
has been a persistent effort by risk managers and monitoring experts to share information with
the public. The issue hence arises of understanding what may have led to the reported
"communication crisis". We propose here to compare the information delivered by the main actors
of monitoring and risk management to Mayotte's inhabitants with what is known of at-risk
populations information needs.
First, we provide a brief overview of what is known about Mayotte's geological setting and
the ongoing seismo-volcanic activity (section 2). We then relate some elements of the political
and social context that contributed to transform a telluric phenomenon with relatively minor
consequence into a situation of crisis (section 3). The corpus and methodology used in our
analysis are described in Section 4. Section 5 describes the successive stages of organisation of
the monitoring and risk management response. As no emergency planning or monitoring had
been done in the department of Mayotte with respect to volcanic issues before May 2018, the
framing of the official response has evolved significantly over time. Documenting this evolution

---

[1] As emphasized by Jasanoff (2004), although science is produced by a specific method in a specific social context, it is influenced by the broader social and political context in which scientists themselves are embedded (this is especially true in risk management contexts when scientists intervene not as researchers but as experts). And science in turn influences the way societies order themselves and organize their response.

was a significant part of our work. It led us to distinguish four main phases (1, 2, 3, 4) that are
presented chronologically in section 5. Because public information strategies have not always
evolved coincidently with monitoring and risk management frameworks, communication issues
are discussed separately in section 6. Analysis of the volume, timing and content of the written
documents used by authorities and scientists to share information with the public leads us to
distinguish three main phases of communication (A, B, C). In section 7, we discuss our results
and issue recommendations to improve future communication strategies. We believe that the
lessons learnt from the relatively long-lasting case study of Mayotte (3 years), in a relatively
unprecedented context (mostly submarine phenomena, leading to "invisible" risks, whose study
requires significant resources and technical innovation), can usefully nourish the reflection carried
out in the literature about risk communication and, more generally, disaster risk reduction.

# 2. Mayotte's geological setting and what is known today about the ongoing seismo-volcanic activity


Mayotte belongs to the Comoros archipelago, a chain of four main volcanic islands that
extends ~E-W between the east African coast and the northern tip of Madagascar (Figure 1).
Recent studies link the formation of these islands to an E-W zone of diffuse transtensional right-
lateral shear at the immature boundary between the Somalia and Lwandle plates (e.g. Famin et
al. 2020, Feuillet et al. 2021, Tzevahirtzian et al. 2021). Following this interpretation, the Comoros
volcanism occurs along en échelon NW-SE tensional fractures affecting the lithosphere in a
context of NE-SW extension (Famin et al., 2020; Feuillet et al., 2021). The location and genesis
of this volcanism would be mostly due to lithospheric deformation (Michon, 2016; Famin et al.,
2020; Feuillet et al., 2021; Tzevahirtzian et al., 2021) rather than to an hotspot trail as previously
proposed by several authors (e.g. Emerick and Duncan, 1982; Class et al., 2009). Volcanism and
formation of the Comoros islands started at least ~10 Ma ago (e.g. Emerick and Duncan, 1982;
Michon, 2016). The Karthala volcano in the westernmost island of Grande Comore (Bachéléry et
al., 2016) is still active today. It is monitored by the Karthala Observatory of the CNDRS (Centre
National de Documentation et de Recherche Scientifique, in Moroni) in collaboration with the
Institut de Physique du Globe in Paris and the University of La Réunion. In Mayotte, recent
volcanism is documented with eruptive products as young as ~4 ky inland (e.g. Pelleter et al.,
2014), and actual at the "new volcanic edifice" (NVE) discovered in May 2018 (Feuillet et al. 2021).
Recent analysis of seismic receiver functions by Dofal et al. (2021) points to a thinned continental
crust beneath Mayotte with a former continental moho at 17-19km depth, underlined by a 9-10km
fast layer interpreted to result from magmatic underplating (Dofal et al., 2021). According to these
authors, the magmatic reservoir feeding Mayotte's new volcanic edifice would be located below
the interface between the underplated magmatic layer and the underlying mantle lithosphere.

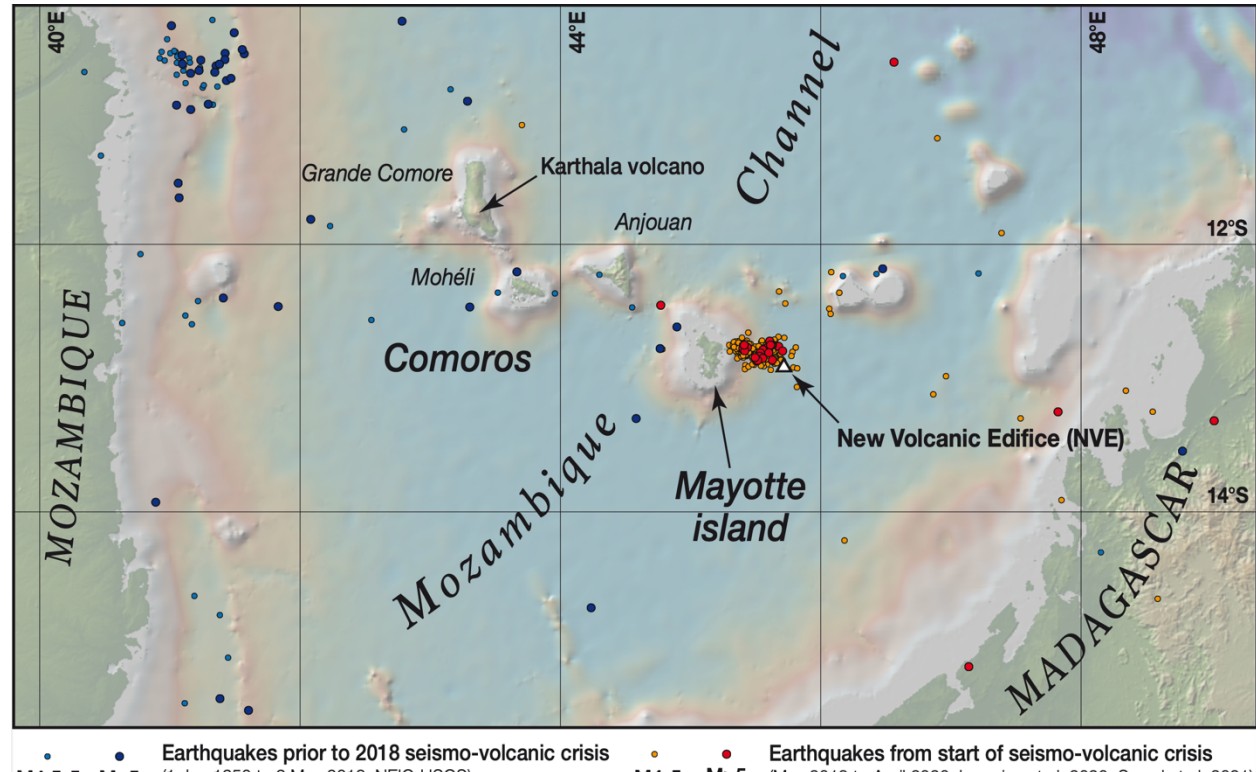

**Figure 1.** Location of Mayotte, easternmost island of the Comoros archipelago. Blue dots: epicenters of seismic events prior to seismic crisis that started on 10 May 2018 (Magnitude ≥4.5, Jan. 1950 to 9 May 2018, USGS catalog); Red (magnitude ≥5) and orange (4 ≤ magnitude < 5) dots show earthquake epicenters with well-constrained hypocentral depth from 10 May 2018 to April 2020 - locations from Lemoine et al. (2020) between May 2018 and March 2019 and REVOSIMA catalog between April 2019 and April 2020 (Saurel et al., 2021). Most earthquakes of the ongoing seismic crisis as well as the new offshore volcanic edifice discovered in May 2019 (Feuillet et al., 2019, 2021) are located 10-50km east of Mayotte island. To avoid problems with mislocated events on this map we excluded epicenters with 10km fixed depth, and only plotted the ones with well-determined hypocentral depths. Topographic and bathymetric visualisation is from GeoMapApp (www.geomapapp.org - CC-BY).

The ongoing activity started on the night of 10 to 11 May 2018 with an earthquake of magnitude ML4.3 felt by the population. Seismicity intensified on 15 May 2018 with several earthquakes of magnitude > 4, all largely felt, and an event of magnitude ML5.8 (MW 5.9) (Lemoine et al., 2020). Although diminishing over time, seismic activity has continued since and is still active at the time of writing, >3.5 years after its beginning. Prior to May 2018, regional instrumental seismicity near the islands (blue dots in Figure 1) was moderate, with the largest magnitudes recorded between $M_b$ 5 and 5.5.

In May 2018, Mayotte's area was poorly instrumented. The ability to identify and precisely locate the earthquakes improved gradually with the development of the network of seismic stations (Bertil et al., 2021; Saurel et al. 2021). The inclusion of underwater stations (OBS for Ocean Bottom Seismometer) from February 2019 (Feuillet et al., 2021, Saurel et al., 2021), and the use of refined seismic velocity models (Lavayssière et al., 2021; Saurel et al., 2021), were

determinant to this respect. The study of the seismicity since the OBS deployment allowed to
locate two clusters of seismicity: a dense "proximal cluster" located close to Mayotte's eastern
coast, and a "distal cluster" located about 30 to 40km east of the islands extending eastward in
the direction of the new volcanic edifice (Feuillet et al. 2021, Saurel et al. 2021, Lavayssière et al.
2021). According to Lemoine et al. (2020), these two clusters are active since the end of June
2018, while, from May to June 2018, the earthquakes occurred in a more distal cluster, shallower
and closer to the new volcanic edifice. This earlier cluster would have included the large
earthquakes that marked the beginning of the crisis. Distal clusters are interpreted to result from
the fracturation and diking processes that allowed magma migration from the deep magma
chamber to the new volcanic edifice (e.g., Cesca et al., 2020; Lemoine et al., 2020; Feuillet et al.,
2021; Lavayssière et al., 2021). The proximal cluster is composed of deep (~35-50km) seismic
events that might be linked to the deformation induced by a deflating deep reservoir (e.g., Feuillet
et al. 2021, Lavayssière et al. 2021). It also contains less deep events (20-35km) that might be
due to stress perturbations around a shallower (~25km) reservoir, as suggested by the location
of very long period seismic events (Feuillet et al. 2021, Lavayssière et al. 2021). Being close to
the islands, it is this proximal seismic cluster, and the magmatic processes related to it, and their
uncertain evolution, that present the real significant hazard.
Inhabitants have mainly experienced the ongoing activity through felt earthquakes. More
than 20 earthquakes with magnitudes 5+ were recorded during the first month of the crisis, from
10 May to mid-June 2018 (Bertil et al., 2021), while ~1900 events with magnitudes >3.5 happened
during the first year (Cesca et al., 2020; Lemoine et al., 2020). About 140 of these earthquakes
were reported as felt by the population in the LastQuake crowdsource-based information app of
the Euro-mediterranean Seismological Center (EMSC-CSEM, 2021). There was a sharp
decrease in the number of felt earthquakes after June 2018, in line with the decrease in the
number of instrumentally recorded earthquakes and of their average magnitude (e.g. Lemoine et
al. 2020, Bertil et al. 2021). EMSC-CSEM catalog reports only ~4 felt events per month until the
end of 2018, and then a moderate recovery in the number of felt events between February and
June 2019 (~9 felt events per month on the average) (the red curve in Figure 3 summarizes this
information).
# 3.    The social and political context of
Mayotte's seismo-volcanic "crisis"
Geoscientists are accustomed to speaking of seismic-volcanic "crises," although the use
of the term "crisis" is not always relevant to disaster risk management definitions. However, in the
case of Mayotte, the observed activity did indeed give rise, at least in the first months, to a crisis
situation that required the intervention of the authorities in charge of civil protection and crisis
management. We relate here some elements of the political and social context that contributed
to this.

● **A vulnerable territory**

Mayotte, which became a French Department in 2011, is a particularly vulnerable territory.
It is marked by great poverty and high social inequality (Roinsard, 2014). In a population of 256
000, 77% live under the poverty line and over 30% are unemployed, 48% are foreign (and often
undocumented), 30% have no access to clean drinking water, and four in ten live in informal
housing (Données 2017 – INSEE, 2021). Mayotte's multiculturalism is a wealth that proves
difficult to manage when the situation requires informing the widest possible audience: 95% of
the population is Muslim (ministère des Outre-mer, 2016), 45% is from the Comoros (INSEE,
2021), and while French remains the official language, about 37% of the population do not speak
it (Données 2017 – INSEE, 2017). Oral culture is the dominant one and the most commonly
spoken languages are Shimaore and Shibushi. There is no real integration between the traditional
culture of the villages and the more westernized culture of large cities (Lambek, 2018). According
to Regnault (2011), "three quarters of the *Mahorais* - rural or, at least, still very attached to their
village - live a culture other than the "westernized" culture of the cities" (*trad. by the authors*). The
relationship with state authorities is complicated by the island's colonial past, but also by a sense
of disappointment among the population, who expected more rapid changes to bring the island
up to French standards after departmentalization (Roinsard, 2019). Since 2011, Mayotte has been
regularly shaken by social crises. The most recent one, which brought the economy to a standstill
for two months in the spring of 2018, was just ending as the first earthquakes began (Roinsard,
2019; Mori, 2021). Lastly, the absence in living memory of seismic and volcanic events in Mayotte
meant that part of the inhabitants were relatively naïve about such risks (although people coming
from the neighboring Comoros islands may have experienced previous seismic and volcanic
crises as four eruptions occurred in 2005, 2006 and 2007, see Morin *et al.*, 2016).

● **A recurring complaint about a lack of information**

The intensity and duration of the initial seismic crisis surprised not only the population but
also the authorities and scientists. On 16 May 2018, the director of the scientific institution locally
in charge of seismic monitoring (the *Bureau de Recherche Géologique et Minière, BRGM*[2])
qualified the activity as "*exceptional beyond anything recorded in Mayotte*" (AFP dispatch picked
up by many media, e.g. Le Point (2018), 16 May 2018). A few days later, the prefect of Mayotte[3]
talked about "*an abnormal and persisting activity*" (*Le Journal de Mayotte*, 19 May 2018). A month
later, in an interview given to the French national press, the director of BRGM Mayotte declared:
*"Unfortunately, we are in the unknown"* (15 June, Le Figaro, 2018b).
Although the earthquakes were of moderate intensity, they affected vulnerable buildings
and their multiplication caused the appearance of cracks leading some municipalities to close
schools (Sira et al., 2018). Local observers reported strong anxiety among inhabitants, many
people leaving their houses to sleep outside (Mori, 2021, Fallou & Bossu, 2019; Fallou et al.,
2020; it was also currently reported in our interviews). They also testified of a general feeling of

---

[2] The Bureau de Recherche Géologique et Minière (BRGM) is a public industrial and commercial institution dedicated to geological resources and placed under the joint supervision of the ministries in charge of ecology, research and economy. It is the only expert earth-sciences institution with a local branch in Mayotte. It is in charge of seismic monitoring in the area when the current crisis begins.
[3] In France, each department is governed by a prefect, appointed by the president. The prefect is responsible for risk and crisis management at the departmental level in coordination with the mayors, who are responsible for risk and crisis management in their municipalities.

confusion linked to the unfamiliar nature of the hazard, and to a lack of information. A group of
citizens created a Facebook feed called "Signalement tremblement de terre de Mayotte" (STTM),
aimed at reporting felt events and at sharing experiences. The success of the feed, which soon
gathered more than 10,000 members (about 4% of the population), attested to the existing thirst
for information. The posts exchanged at that time show a lack of confidence in the authorities'
willingness to take charge of the situation: *"Earthquakes that sometimes exceed magnitude 5,*
*cracks in buildings, fires, landslides, etc.... and no real reaction from the state apart from*
*information on the magnitude of the tremors already felt."* (excerpt from STTM Facebook group,
26 May 2018); "*How much do you want to bet that in a year nothing will have been done? As soon*
*as the crisis passes we[4] play the watch hoping that the next one will come when we leave the*
*island. That's how the administration has managed Mayotte for decades.*" (excerpt from STTM
Facebook group, 27 May 2018). On 5 June 2018, the deputy of Mayotte in the French national
assembly warned the government against the consequences of a lack of public information
leading to the spread of *"false information fueled by fantasies that have the effect of increasing*
*people's anxiety, generating a state of panic and even psychosis"* (Ali, 2018). Eight months later,
in February 2019, members of the STTM facebook feed published an open letter urging the state,
local elected representatives and scientists to provide more information about the ongoing activity
(Picard, 2019). Although this group is not really representative of the sociology or the demography
of Mayotte's population, it soon became a serious interlocutor for the local authorities, and the
prefect invited its most visible members to the discussion table in 2019 (Journal de Mayotte, 9
August, YD, 2019). It remains today one of the public arenas where information about the seismic-
volcanic crisis is followed with the most attention.
It took a whole year between the beginning of the seismic crisis and the official declaration,
in an interministerial press release dated from 16 May 2019 (*ministère de la Transition écologique*
*et solidaire, ministère de l'Enseignement supérieur, de la recherche et de l'innovation, ministère*
*des Outre-mer, ministère de l'Intérieur*, 2019), of the discovery of the new volcanic edifice. The
event closed a year of questioning about the possible origin of earthquakes. The unexpected
*"birth of a new volcano"* (BBC - Science in Action, 2019) caused enthusiasm in the national and
international scientific community, and in the media (e.g., Andrews, 2019; Minassian, 2019; Wei-
Haas, 2019; Devès et al., 2022). The discovery has been described as "exceptional": first,
because of the large volume of lavas involved, more than $5\,km^3$ (Feuillet et al., 2021) -
corresponding to the largest eruption ever observed with modern techniques (Cesca et al., 2020;
Feuillet et al., 2021; Thordarson & Self, 1993) - and, second, because of the submarine nature of
the activity - marking the beginning of an exciting scientific adventure to develop new techniques
of observation. The local press welcomed this sudden interest in Mayotte's actuality (Devès et al.,
2022), the volcano being presented as a more positive way of talking about the *101st department*
than the usual references to its social misery (*Journal de Mayotte*, 28 May 2018). But "discovering"
the volcano is insufficient to characterize the associated threats. In this sense, the advance in
knowledge showed itself to be frustrating for the inhabitants, for the authorities, and for journalists
alike (Devès et al., 2022). In June 2019, STTM's facebook feed members were still complaining
about the official communication: *"Say nothing, explain nothing... Can only create confusion...*
*Questions that go around in circles because we don't have the answers! When there is neither*

---

[4] "We" refers here to the civil servants coming from metropolitan France to work in the overseas department of Mayotte.

*answer nor explanation ... One can only wonder ... Why this? What interest or motivation do they have in not giving the information ... They would like the population to worry: they couldn't do better! The sickly inability of administrations to communicate …"* (excerpt from Facebook group STTM, 20 June 2019).

# 4.  Material and methods

The present research is part of a research project entitled MAY'VOLCANO dedicated to the study of the circulation of knowledge between scientists, risk and crisis management actors, the media and the population of Mayotte during the current seismo-volcanic crisis. This paper aims at providing a first analytical view of the public information process, and of its potential limitations.

The empirical data for the research presented here were collected between 10 May 2018 and 1 April 2021, covering more or less the three first years of the ongoing seismo-volcanic "crisis". The work was organized in three tasks: 1) documenting and understanding the organisation of the monitoring and risk management response and its evolution over time, 2) documenting and understanding the organisation of the process of public information and its evolution over time, and 3) examining the process of public information with regard to what is known of at-risk population information needs. The first two tasks were done in parallel. In the following, we describe the empirical data and the methods used to complete each of these tasks. The corresponding results are presented in section 5 (task 1), 6 (task 2) and 7 (task 3).

### 4.1. Documenting and understanding the organisation of the "official response" and its evolution over time

Our first task was to capture and understand the organisation of the "official response". By "official response", we mean the decisions and actions taken by the local and national authorities in charge of risk and crisis management and by the scientific experts in charge of monitoring the ongoing seismo-volcanic activity. As emphasized in the introduction, the framing of that response evolved significantly over time and it was important to be able to document and describe these evolutions before addressing the issue of public information.

The methods chosen were participant observation, semi-structured interviews, collection and analysis of written archives. The fact that three of the authors worked at the Institut de Physique du Globe de Paris (IPGP), which is currently in charge of monitoring the activity, facilitated contact with experts. The involvement of the first author in previous research projects associating crisis management officials facilitated contact with authorities.

Participant observation was done within the framework of a day-to-day cohabitation with scientists at IPGP, within the scientific council of the REVOSIMA since February 2020 (when the first author was invited to join) and, between January and June 2021, within a working group coordinated by the interministerial delegation for major risk reduction in overseas territories (the

Délégation interministérielle aux Risques majeurs en Outre-mer, DIRMOM) who developed a
sensibilisation campaign (using videos) about the seismic and tsunami risks in Mayotte.
15 semi-structured interviews were conducted with the persons who were identified as pivotal
to the overall monitoring and risk/crisis management process: 8 with scientists directly involved in
the organisation of monitoring (sometimes at different moments of the crisis), 7 with risk or crisis
managers acting at the local, national or inter ministerial levels. Two of these persons were
interviewed twice, before and after the creation of the REVOSIMA which allowed us to gain a
better insight into the associated changes. Most interviews were conducted via visioconference
because of the restrictions due to the COVID-19 pandemy. During the interviews, we asked
questions about the actors involved in monitoring, risk and crisis management, about their role,
about the procedures, contents and formats used to exchange information, between them, with
the media and the public. We also asked more specific questions about the communication
process (see section 4.2). All interviews were recorded (with the agreement of the interviewees)
and transcribed soon after. The transcriptions were anonymized when used for discussion
between the members of the team (only the first author has access to the original files as she was
the one conducting the interviews). Citations taken from interviews for illustration in the present
paper are anonymized to respect interviewees' confidentiality. We also provide our own English
translation. The interviews were analyzed qualitatively with the aim to understand the organisation
of the official response and its evolution. The chosen method places emphasis on the meaning
rather than the quantification of the materials.
Regarding the collection of archives, we collected public press releases, public scientific
bulletins and official reports. Interviewees often spontaneously shared the materials they used to
communicate and the materials on which they based their decision, such as internal notes and
reports. We cite here only the documents that are public.
The work carried out on the basis of those data allowed us to identify the main actors to be
considered for studying the process of public information (Figure 2).

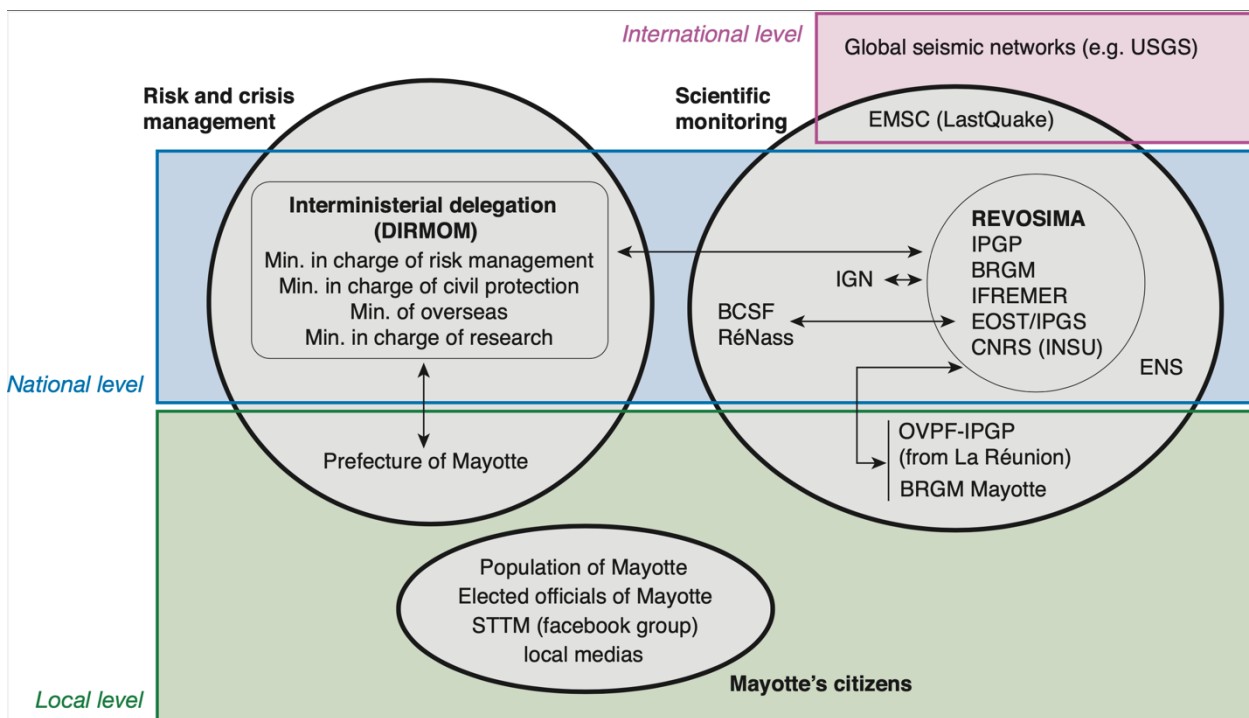

**Figure 2:** Cartography of the actors who played an active role in public information about the seismo-
volcanic crisis of Mayotte during our period of study.

Two main categories of actors are distinguished according to their function: risk and crisis
management or scientific monitoring.
On the risk and crisis management side, the main actors are 1) the *p*refecture of Mayotte,
which is the body representing and implementing government policy at the local level, and 2) the
ministries concerned with risk prevention (ministère de la Transition écologique et solidaire), civil
protection (ministère de l'Intérieur), research (ministère de l'Enseignement supérieur, de la
recherche et de l'innovation), and overseas administration (ministère des Outre-mer). The
interministerial level is also to be considered because of the active role played by a temporary
interministerial delegation called DIRMOM (*Délégation interministérielle aux Risques majeurs en*
*Outre-mer*) whose task was to improve coordination between ministries on the topic of major risk
reduction in the French overseas. The delegation was in activity between April 2019 and June
2021. The end of our study period therefore corresponds approximately to the end of the
DIRMOM's activity, at the dawn of a possible reorganisation of interministerial coordination on
major risk management overseas. In the French system, mayors are usually key actors of risk
and crisis management. But, in the case of the seismo-volcanic crisis of Mayotte, it soon appeared
that public information was mainly being orchestrated at the departemental and national levels
(anonymous from interviews conducted in June 2020, April, June and September 2021). The
explanation that was given to us by interviewees is that the initial crisis overwhelmed the capacity
of response of local mayors requiring the intervention of the prefecture of Mayotte, with the
support of the national level.
On the monitoring side, the number of actors involved has evolved significantly over time.
In summary[5], the Institut de Physique du Globe de Paris (IPGP), the School and Observatory of
Earth Sciences in relation with the École et observatoire des sciences de la terre / Institut de
Physique du Globe de Strasbourg (hereafter referred as EOST), the Bureau de Recherche
Géologique et Minière (BRGM) and the Institut Français de Recherche pour l'Exploitation de la
Mer (IFREMER) have been directly involved in monitoring, although in different ways over time.
They are the main partners of the REVOSIMA network. The latter, born in June 2019, is operated
by the IPGP from its closest observatory of the Indian Ocean region, i.e. the Observatoire
volcanologique du Piton de la Fournaise (OVPF) in Reunion Island, and with the support of the
antenna of BRGM in Mayotte. The Bureau central sismologique français - Réseau national de
surveillance sismique (BCSF-RéNass), the European-Mediterranean Seismological Centre
(EMSC) and the National Institute of Geographic and Forest Information (IGN) centralise,
distribute or provide data.

---

- [5] The Bureau de Recherche Géologique et Minière (BRGM) and the Institut Français de Recherche pour l'Exploitation de la Mer (IFREMER) are public industrial and commercial institutions dedicated to, respectively, georessources and marine resources placed under the joint authority of the Ministries in charge of ecology, research and, respectively, economy or agronomy. The National Institute of Geographic and Forest Information (IGN) is a public administrative establishment placed under the joint authority of the Ministries in charge of ecology and forestry.

- The Institut de Physique du Globe de Paris (IPGP) is an institution for higher education and research in geosciences which is in charge of certified observation services in volcanology, and seismology through its permanent volcanological and seismological observatories like the one in La Réunion island (OVPF for Observatoire Volcanologique du Piton de la Fournaise). It operates the Volcanological and Seismological Monitoring Network of Mayotte (REVOSIMA).

- The School and Observatory of Earth Sciences (EOST) is an institution under the supervisory authority of the University of Strasbourg and the CNRS (French National Center for Scientific Research) in charge of education, research, and observation in Earth Science. The IPGP and EOST equip and maintain global geophysics networks that monitor seismic activity (GEOSCOPE network) around the globe. EOST is sometimes referred to as the Institut de physique du Globe de Strasbourg (IPGS), the two bodies having intimate links. The EOST pilots the BCSF-RéNass, Bureau central sismologique français - Réseau national de surveillance sismique, which is in charge of centralising, archiving and distributing national seismic data. The BCSF-RéNass issues a bulletin after each event and collects public testimonies of felt earthquakes (www.franceseisme.fr). It also provides assistance to the public authorities by sending a task force of seismologists (GIM for Groupe d'intervention macrosismique) to estimate impacts after significant earthquakes in French territories.

- The French National Centre for Scientific Research (CNRS) is an interdisciplinary public research organisation under the administrative supervision of the French Ministry of Higher Education and Research. A significant part of French researchers belong to CNRS and work within laboratories which are placed under the joint authorities of the CNRS and the local university. The National Institute for Universe Sciences from CNRS (INSU) has the mission to develop and coordinate French research in astronomy and Earth sciences, as well as ocean, atmospheric, and space sciences.

- The European-Mediterranean Seismological Centre (EMSC) runs an Earthquake Alert System for potentially damaging earthquakes in the Euro-Mediteranean region. As BCSF-RéNass, EMSC collects testimonies through its Lastquake application (e.g., Bossu et al., 2019). Within the hour following the occurrence of an earthquake, EMSC publishes a web page with its epicentre and magnitude, and the collected testimonies.

## 4.2. Documenting and understanding the organisation of the process of public information and its evolution over time

The ultimate goal of this research being to examine the process of public information, it required documenting and understanding how the above-mentioned network of actors organized its "external" communication (Becker et al., 2018) and how it evolved with time. We used the same methods as those mentioned in section 4.1. In addition to the questions listed earlier, we also asked the interviewees what were the role of the various actors with respect to public information, what role they played at an individual scale, what were the most important moments for them with respect to public information and to give their view on the effectiveness of that information regarding risk reduction. We also took note of the media most commonly used to share information with the public and decided to systematically collect the documents that were available (either online or with the help of the interviewees).

We searched the archives and in particular the web archives of the scientific and state institutions involved in monitoring and risk management. We collected all the written documents. By the end of our period of study, we had collected 320 items including press releases, scientific bulletins, news on websites and public notes (Table 1, a table listing all the documents we collected during our period of study is provided in supplementary information). Hereafter, we are citing scientific bulletins and websites as references (including their URL when existing) while authorities' press releases are given in the supplementary dataset (press releases are typically from the prefecture of Mayotte but there are also a few press releases from the government and from ministries). We did not consider the numerous automatic bulletins emitted by REVOSIMA (daily automatic bulletins are emitted since march 2020), BCSF-RéNass and EMSC but we included the report published by the BCSF-RéNass's Groupe d'intervention macrosismique (GIM) and a web article from the EMSC aiming at providing a global view of the seismic crisis. We also included in our database the five academic papers (one was a preprint version of a submitted paper) dedicated to the crisis that were published during our period of study (Cesca et al., 2020; Famin et al., 2020; Feuillet et al., 2021; Lemoine et al., 2020; Tzevahirtzian et al., 2021) and commented by the press and/or the members of STTM facebook group. We also took into account the contribution of individual researchers who issued key analyses at crucial times during the crisis (Briole, 2018).

Each item was downloaded, stored in pdf under a specific ID, and then read independently by 2 to 3 researchers who completed a table with information about format and content. Disagreements were discussed and solved collectively. We took note of the ID, the date of publication, the URL (when existing), the publishing authors/institutions, the title, the public it aimed to, the number of words, the presence or absence of illustrations and the nature of these illustrations (scientific, local, etc.). We also took note of the main topics covered by the text and of the list of actors that were mentioned. This dataset was used to quantify the volume and timing of public information, and to undertake a qualitative analysis of content.

To complete our understanding of the public information process, we also explored Facebook publication feeds when they existed (i.e. for OVPF-IPGP, REVOSIMA and prefecture of Mayotte) but without aiming for exhaustiveness as it was difficult to achieve without adequate tools.

Using the catalog of felt seismicity provided by EMSC (EMSC-CSEM, 2021), we compared
the publication rate to the number of earthquakes felt by Mayotte citizens and its evolution in time
(Figures 3 and 4). This allowed us to put the scientist's and authorities' communication effort in
perspective with the evolution of the geophysical signal that directly affected the population.
**4.3. Examining the process of public information with regard to what is known of at-**
**risk population information needs**
The combination of these data (archives, interviews, notes of participant observation,
written documents used by the actors to share information with the public) provided the basis for
examining the public information process with regard to what is known of at-risk populations'
information needs. The latter is inferred from the existing literature on risk communication (which
is abundant on this particular topic, see section 7), while bearing in mind the social and cultural
context of Mayotte.
We also explored STTM's Facebook publication feed but, again, without aiming for
exhaustiveness as it was difficult to achieve without adequate tools. Hereafter, we use excerpts
from STTM facebook posts to illustrate some of our statements. We anonymised these citations,
and provide our own English translation (anonymised French original versions of the facebook
posts are given in supplementary dataset).
# 5. The organisation of the "official response"
and its evolution
As no emergency planning or monitoring had ever been done in the department of Mayotte
with respect to volcanic issues before May 2018, the framing of the official response has evolved
significantly over time. Here we provide a description of its gradual organisation. We distinguish
four main successive phases (1, 2, 3, 4). The first phase goes from the recording of the first
earthquakes to the recording of the first unambiguous signals of a volcanic component. The
second phase corresponds to the mobilization of scientists, and funding agencies in relation to
ministries, to get the financial means to instrument the area. The third phase runs from the first
measurement campaigns to the proof of the volcanic activity which signed the official setting up
of the seismo-volcanic monitoring network of REVOSIMA. The fourth phase begins with the
official creation of REVOSIMA and ends with our windows of study. Figure 3 summarizes the key
events that marked each of these four phases. In addition to the events linked to monitoring, we
also discuss some key events in the response of scientists, authorities and inhabitants of Mayotte.


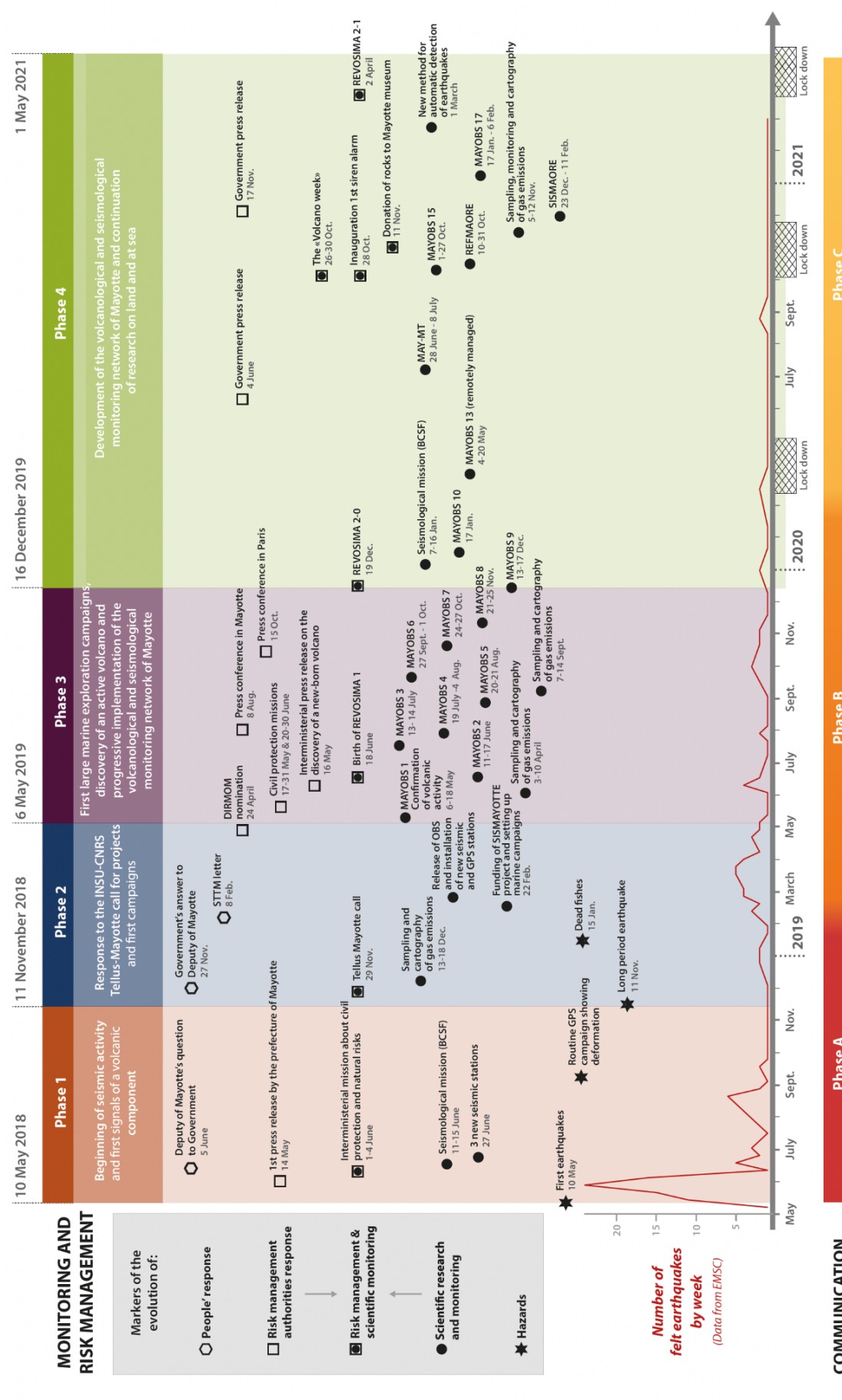

**Figure 3:** Major phases and markers of the response by local and national authorities in charge of risk and
crisis management and by scientific experts in charge of monitoring the seismic-volcanic activity in
Mayotte. Our period of study extends from 10 May 2018 to 1 April 2021. The lockdown periods that are
shown are those of metropolitan France (note that most of the scientific institutions involved in
monitoring are located in metropolitan France). Mayotte endured longer lockdowns in spring 2020 and
2021 but there was no proper lock down in autumn 2020.

● **Phase 1: 10 May 2018 to 10 November 2018**
During the **first phase of the crisis**, the French Geological Bureau (BRGM) played a
central role. It was the only geo-scientific institution with a permanent office in Mayotte and, at the
beginning of the seismic crisis, it was in charge of maintaining the only 3 accelerometric seismic
stations installed on the island (known as moderately active). BRGM Mayotte was hence the
natural interlocutor of the local and national authorities for decision support. But the situation was
difficult as crucial data were missing. Only the largest magnitude earthquakes (M>5) were
reported by global seismic networks while the existing local network – the few accelerometric
stations in Mayotte completed by few regional stations in Comoros and in Madagascar – did not
allow a good record of the surge of moderate magnitude earthquakes felt by the population.
Because of this inadequate network, the BRGM operators initially encountered difficulties in
accurately locating the earthquakes and assessing their epicentral depths (see section 2).
In June 2018, the persistence of the seismic crisis led to the involvement of new actors.
Ministries in charge of civil protection (ministère de l'Intérieur) and disaster risk prevention
(ministère de la Transition écologique et solidaire) sent an interministerial mission composed of
civil protection experts and seismologists (e.g., Mayotte la 1ère, 2018; Perzo, 2018b). The experts
concluded that the impact of the earthquakes mainly resulted in an aggravation of disorders on
buildings that were already vulnerable (widening, elongation of cracks) and reported that about
thirty people got minor injuries that were indirectly linked with the earthquakes (e.g. falling down
stairs to get out of the house). They also outlined that the repetition of shaking had been causing
a feeling of anxiety and fear among the population, all the more marked as this seismic swarm
phenomenon was unknown in Mayotte until then[6]. Mid-June 2018, a team of seismologists from
BCSF-RéNass was sent to *"estimate the levels of damage induced by this seismic swarm*
*according to the vulnerability of the buildings at the date of the field analysis"* (Sira et al., 2018).
3 more seismic stations were installed (two short-period RaspberryShake velocimeters by the
BCSF, one broad-band velocimeter in the frame of the 'Sismo à l'École' network). During the
summer, scientists from IPGP and EOST helped the BRGM team to monitor the activity[7]. In July,
the French scientific community started organising to seek funding to instrument the area, notably
at sea. A note was sent to the French National Centre for Scientific Research (CNRS) to attract
funding agencies' attention to Mayotte's issues[8].

---

[6] The problem of anxiety was addressed with the opening of a toll-free phone number and a psychological support unit
at the local hospital (Press release of the prefecture of Mayotte, 19 June 2018)
[7] Until the creation of REVOSIMA, real-time data processing was organized through the voluntary commitment of
scientists.
[8] The issue of funding is not simple. The activity being mostly submarine, surveys have to be done mostly offshore
using research vessels and heavy human and technical logistics. The funding to be mobilized is typically of the order
of several million euros per year. In parallel, one also has to deal with vessel's availability for their work programs are
often planned years in advance. However, several scientists we interviewed claim that the rapid mobilization of fifty

In September, routine satellite measurements (using Global Navigation Satellite System,
GNSS) led by the IGN revealed strong displacement anomalies affecting stations located on the
island. Researchers from the Ecole Normale Supérieure (ENS) Geoscience Lab. analyzed the
data, tracing the onset of surface deformation back to July 2018 (Briole, 2018). They explained it
by the deflation of a huge magmatic chamber located off the coast of Mayotte. The lack of
geological observations offshore Mayotte was still preventing a good understanding of the
phenomenon but the scientific community urged public authorities to fund geophysical
instrumentation and surveys in the region.
● **Phase 2: 11 November 2018 to 5 May 2019**
The **second phase of the crisis** started on 11 November 2018 with a long period
earthquake with peculiar characteristics (a very long trend of monochromatic seismic waves, e.g.,
Cesca et al. 2020, Lemoine et al. 2020). The event, not felt by the population because of its long
period character, was recorded by global seismic networks. It was much discussed on social
networks and appeared to be mentioned in the international and soon national and local press
(see discussion in Lacassin et al., 2020). It supported the volcanic hypothesis (Cesca et al., 2020;
Lemoine et al., 2020). Mid-november, a meeting was organised with representatives of the four
ministries, scientists and scientific institutional stakeholders like CNRS-INSU. On 29 November,
public authorities set up a call for projects to fund observation and research in the area. The call,
named *"Tellus-Mayotte"*, was coordinated by the CNRS-INSU and co-financed by the ministry in
charge of disaster risk prevention (*ministère de la Transition écologique et solidaire*).
In January 2019, fishermen reported dead deep sea fishes at the surface of the ocean
east of Mayotte (Perzo, 2019a)[9]. On 22 January, three projects were eventually selected on the
Tellus Mayotte call, involving 11 laboratories and 44 scientists from CNRS, IPGP, EOST, BRGM,
Ifremer and IGN. On 22 February, CNRS, IPGP, BRGM and EOST announced the launch of the
first major monitoring missions. Between February and March 2019, 6 OBSs were deployed at
sea in the frame of these Tellus-Mayotte projects, and new seismic and GNSS stations were
installed on land (by OVPF-IPGP, BRGM, EOST). A team from the University of La Réunion
associated with OVPF-IPGP carried out field missions to consolidate knowledge of the tectonic
and volcanic history of Mayotte.
● **Phase 3: 3 May 2019 to 5 December 2019**
**The third phase of the crisis** started with the first MAYOBS marine campaigns on the
scientific ship *Marion Dufresne* (MAYOBS 1 on 6-18 May 2019 and MAYOBS 2 on 11-17 June).
The campaigns were led under the auspices of the CNRS and involved scientists from BRGM,
IPGP, EOST, IFREMER, the University Clermont Auvergne, the University of La Rochelle with
the support of IGN, the national center for space studies (*Centre national d'études spatiales*,
CNES) and the service hydrographic and oceanographic marine observations (*Service
hydrographique et océanographique de la marine*, SHOM). The OBSs deployed in February were
retrieved and new ones were released. The data allowed relocating the earthquakes and

thousand euros in funding would have provided enough knowledge by the end of summer 2018 to confirm the volcanic
origin of the seismicity. So there is a debate about the agility of the scientific and administrative governance in
organizing the monitoring response as quickly as possible.
[9] It is the first time the existence of dead deep sea fishes were made public.

specifying the location of the seismic swarms (Deplus et al., 2019; Feuillet et al., 2019, 2021;
Jacques et al., 2019; Saurel et al., 2019). Scientists  also acquired high-resolution marine
geophysical data, studied the water column and carried out rock dredging operations on the
seafloor. An ongoing deep sea volcanic activity was discovered with a new ~800m high
underwater volcanic edifice, confirming the already suspected volcanic hypothesis. The discovery
was announced by an official press release signed by four ministries (e.g., *ministère de la*
*Transition écologique et solidaire, ministère de l'Enseignement supérieur de la recherche et de*
*l'innovation, ministère des Outre-Mer, ministère de l'Intérieur,* 2019) and relayed by the scientific
institutions involved in the campaign on their websites.
Numerous other marine campaigns followed, allowing to refine progressively the
understanding of the phenomenon (see Feuillet et al. (2019) to access the MAYOBS campaigns'
reports). On 18 June 2019, an interministerial meeting set up a scientific and technical committee
to monitor the activity and officialized the creation of the Volcanological and Seismological
Monitoring Network of Mayotte (REVOSIMA) with the implementation of *"a monitoring of*
*volcanological and seismological activity in real time and continuously"* (IPGP, 2019b, published
on 27 August 2019, translation by the authors). Several phases were envisaged for the
implementation of this network. In a first phase, the REVOSIMA (called REVOSIMA 1 by the
actors) was supported by a 2.5 million euros fund in order to establish a monitoring network and
to guarantee a scientific follow-up of the phenomenon with the implementation of new oceanic
campaigns aiming at deploying and recovering OBS. The monitoring mission was entrusted to
the IPGP, already in charge of the other French volcanological and seismic observatories. IPGP
decided to operate this network through the Observatoire volcanologique du Piton de la Fournaise
(OVFP-IPGP) in co-responsibility with the BRGM and its regional direction in Mayotte. The
REVOSIMA's mandate was outlined as follows to: *"i) monitor the seismo-eruptive dynamics on*
*land and at sea, in particular in connection with offshore campaigns and underwater*
*instrumentation to monitor the possible migration of seismicity and volcanism, ii) monitor marine*
*deformation and submersion, iii) characterize and monitor gravitational instabilities and tsunami*
*hazard, iv) improve knowledge of the tectonics and geodynamic context of Mayotte, v) monitor*
*the geochemistry of volcanic fluids."* (IPGP, 2019b, published on 27 August 2019, translation by
the authors). In October 2019, a "pickathon" was organised by the REVOSIMA's scientists in
order to speed up the process of seismicity relocation.
● **Phase 4: 16 December 2019 to 1 April 2021**
**The fourth phase of the crisis** corresponds to the progressive development of the
volcanological and seismological monitoring network which allowed the progress of research on
land and at sea (there has been more than eight research and monitoring campaigns since
december 2019). In December 2019, a new interministerial meeting ratified the perpetuation of
the surveillance network and the release of 4.5 million Euros funding. REVOSIMA 2 was launched
at the beginning of 2020. In January 2020, seismologists of BCSF-RéNass came back to Mayotte
to trace the evolution of damages due to the earthquakes from June 2018 and a second pickathon
was organised to relocate seismicity. From March 2020 onwards, the actors had to deal with
disruptions due to the international pandemic of COVID-19. A double maritime campaign
(MAYOBS 13-1, MAYOBS 13-2) was nevertheless organized in May with the support of the
French Navy. The second campaign was remotely operated by scientists from IFREMER, IPGP,
BRGM and CNRS located in metropolitan France. It was followed, in June, by a magnetotelluric
campaign (MAY-MT) and, in October, by a seismic-refraction campaign (REFMAORE), both
coordinated by BRGM. The oceanographic campaigns have continued at a steady pace since
then, despite the second and third COVID-19 lock downs. The only notable change, at the end of
our study period, was the improvement of the automatic earthquake location method announced
by REVOSIMA in March 2021.

# 614    6. The organisation of the process of public
# 615       information and its evolution

Table 1 lists the preferred publication format and the volume of communication issued by
the main actors in charge of monitoring and crisis and risk management during our period of
study. Figure 4 shows that the number and frequency of publications has varied greatly over time
and among actors. Public information was particularly intense during the first six weeks of the
crisis and continued with some regularity throughout 2018. The average number of
communications per day was 6,8 during the first phase of the crisis (phase 1), compared to 1,3
(phase 2), 1,2 (phase 3) and 1,0 (phase 4) during subsequent phases. Over 90% of all press
releases and scientific bulletins issued by authorities and scientists during our period of study are
dated from 2018 i.e., during the period qualified by Fallou et al. (2020) as an "information vacuum".
This finding deserves an in-depth analysis to understand the discrepancy between the initial high
communication rate and the perceived lack of information. Hence, hereafter, we analyze in detail
not only the frequency but also the content and modalities of public information and its evolution
over time. Three main phases are distinguished (A, B, C) that are discussed in relation to the
phases 1, 2, 3, 4 describing the evolution of the monitoring and risk management response
(Figures 3 and 4).

**Table 1.** Format and volume of the documents made public by the main actors of scientific monitoring and
risk and crisis management during our period of study. A table listing all the documents we collected during
our period of study is provided in supplementary information. As discussed in the text, we only count a
report and a web article for, respectively, the BCSF-RéNass and the EMSC, and not their automatic reports.
We do not count the automatic bulletins from REVOSIMA. We include the five academic articles dedicated
to the understanding of the phenomena occurring in Mayotte that were published during our study period.

|  | Scientific bulletins | Press releases | News on website | Public notes | Academic papers | *TOTAL* |
|---|---|---|---|---|---|---|
| ***Scientific monitoring*** |  |  |  |  |  |  |
| BRGM | 104 |  | 22 |  |  | **126** |
| REVOSIMA | 40 | 1 |  |  |  | **41** |
| IPGP |  | 1 | 15 |  |  | **16** |
| IFREMER |  |  | 10 |  |  | **10** |
| Researchers |  |  |  | 4 | 5 | **9** |
| EOST |  |  | 8 |  |  | **8** |
| CNRS/CNRS-INSU |  | 2 | 1 |  |  | **3** |
| IGN |  |  | 1 |  |  | **1** |
| EMSC |  |  | 1 |  |  | **1** |
| BCSF-RéNaSS | 1 |  |  |  |  | **1** |
| ***Risk management*** |  |  |  |  |  |  |
| Prefecture of Mayotte |  | 100 |  |  |  | **100** |
| Ministries/Governement |  | 4 |  |  |  | **4** |
| *TOTAL* | **145** | **108** | **58** | **4** | **5** | **320** |


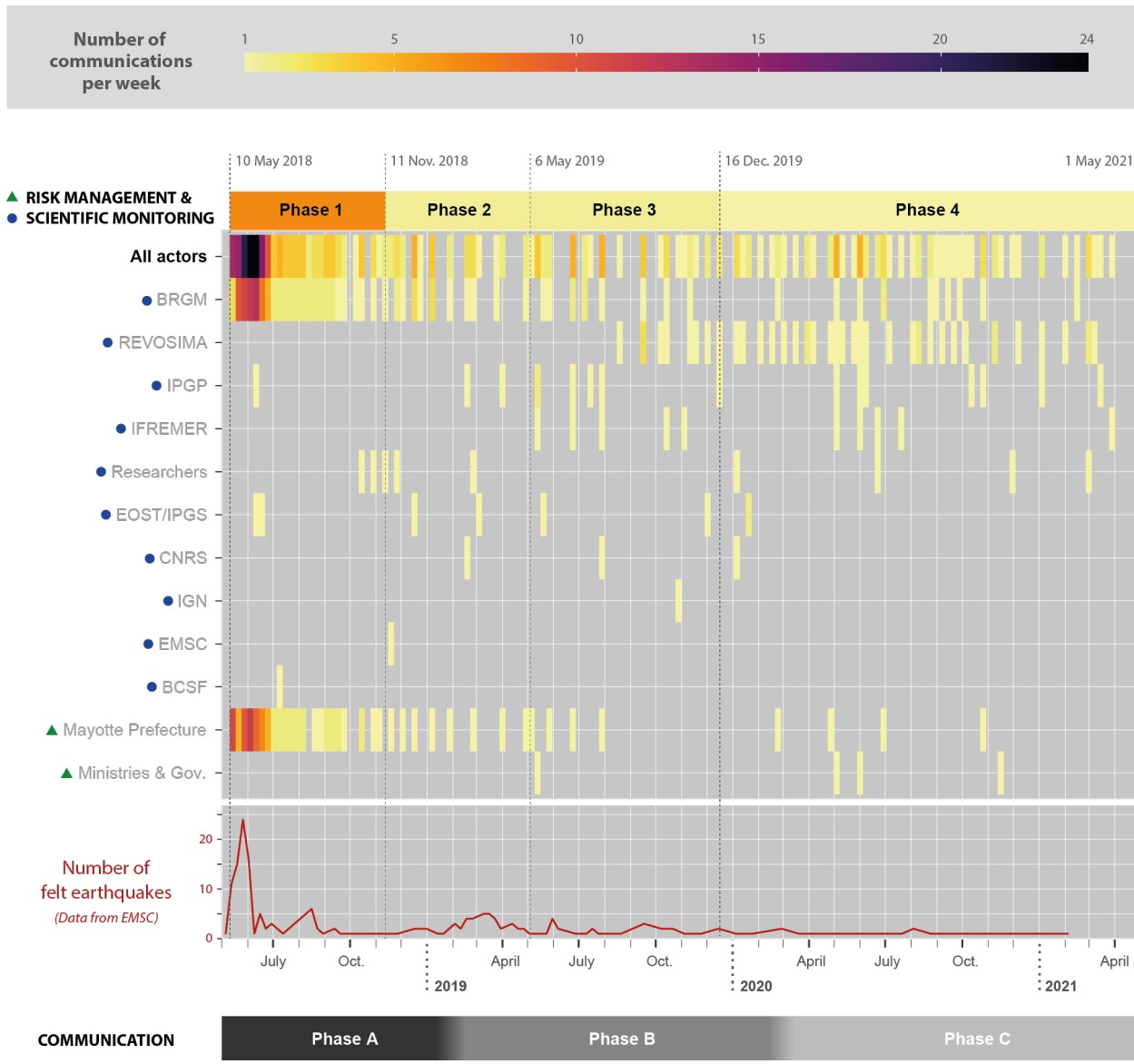

**Figure 4.** Number of documents made public per week by the main actors of monitoring and risk and crisis management. The average number of documents published per day is indicated for each of the phases identified in Figure 3.

- **Phase A: from the beginning of the crisis to February 2019**

Between the beginning of the seismic crisis and February 2019, the modalities of communication did not vary much. The local stakeholders in charge of monitoring and risk and crisis management, BRGM and the prefecture of Mayotte, were the main contributors. Other scientific actors, such as the IPGP and the EOST who were gradually getting involved in monitoring from the first months of the crisis, were only communicating punctually to report on the geodynamic context of the activity and/or on their involvement in the collect and treatment of data: e.g. on 11 June 2018, EOST announced the dispatch of the macroseismic response mission

(GIM) to Mayotte (EOST, 2018a); on 12 June, IPGP published an information brief on the ongoing
crisis in Mayotte (IPGP, 2018).
The first communication to the public was a press release from the prefecture of Mayotte
on 14 May 2018. Referring to the monitoring undertaken by the BRGM since 10 May 2018, it
mentioned a *"swarm of earthquakes"*, distinguished it from seismic aftershocks and recalled the
safety instructions to be followed in case of earthquakes. Three press releases were published
on 15 May that listed the time and magnitude of felt earthquakes and specified that *"all the*
*earthquakes [took] place in the same sector (around 50km off Mayotte) and, although located at*
*sea, [were] too weak to generate a tsunami"*. Confronted with the repetition of felt earthquakes,
the prefect of Mayotte activated a crisis unit on 16 May 2018. From then on, the Prefecture
publishedpress releases on a daily basis (sometimes more) while the BRGM, switching to "crisis
monitoring", published daily reports[10]. As testified by several interviewees, during that first phase
of the crisis, the local branch of BRGM was put under strong pressure *"to be able to inform, almost*
*'day and night', the authorities on the magnitude, on the location of the earthquakes, a more*
*precise location than the one announced by the international networks which were not reliable*
*because of their distance"* (anonymous, interview in May 2020).
During the first weeks of the crisis, the scientific reports and official press releases followed
one another within a few hours. BRGM published its bulletins on the BRGM website[11], while the
prefecture sent press releases to the press and published them on Facebook. These official press
releases generally reproduced the elements communicated by the BRGM. They remained often
very technical, recalling the number of earthquakes recorded per day, their magnitude, the time
at which they were detected and their distance from the island (the reports mentioned
uncertainties of the order of 10-15 km). The prefecture's press releases could contain additional
elements about impacts (injuries, building damage) and often recalled safety instructions. They
also provided information about the decisions taken by the prefecture to support the inhabitants
of the island (e.g. the setting up of a toll-free phone number and the opening of a psychological
support unit ; the demand for (and arrival of) a support mission of civil protection and risk
management in June 2018).
Mid-June 2018, the BRGM published a Frequently Asked Questions (FAQ) on its website
explaining the state of knowledge and the main uncertainties. But, as written a few months later
by the ministry in charge of civil protection (*ministère de l'Intérieur*) in its answer to the deputy of
Mayotte, *"the most inventive explanations have found an echo in part of the population*
*(conspiracy, actions of evil spirits, etc.) and communication is proving difficult. The state has*
*obviously been concerned about this situation since the beginning of the event, and everything*
*possible is being done to inform the population in a reliable manner"* (Question à l'assemblée
nationale n°8992, 27 November 2018, Ali, 2018). Among the incorrect explanations that had
emerged, a popular one was that the earthquakes were caused by oil exploration off the coast of
Mayotte (Fallou et al., 2020; Mori, 2021). The hypothesis of a volcanic cause had also surfaced:

---

[10] https://www.brgm.fr/fr/actualite/dossier-thematique/volcan-seismes-mayotte-brgm-fortement-implique
[11] *id.*

it was discussed on the websites of national scientific laboratories (EOST, 2018b; IPGP, 2018)
and in the local press (e.g., YD, 2018) as early as May-June 2018.

From the end of june 2018, the number of communications decreased with the decrease
in seismic activity (2 BRGM bulletins per week from 29 June 2018). In September 2018, BRGM
announced that *"the swarm [was] still running [but that] the lull observed since the end of June*
*[justified] the change from "crisis" monitoring to "routine" monitoring"* (bulletin of 17 Sept, BRGM,
2018a). From then on, BRGM published bulletins twice a month, with exceptional bulletins in case
of felt earthquakes. In October 2018, analysing the routine GNSS measurements led by the IGN,
a geophysicist from the Ecole Normale Supérieure suggested that the seismicity could be related
to the deflation of a deep magma chamber. These results were published in the form of notes on
the public website of the laboratory in October, November and December 2018 (Briole, 2018). In
the opinion of several scientists we interviewed, the *"wild"* (sic) publication of his results played
an important role in raising awareness of the importance of this seismic crisis among the scientific
community and authorities in charge of risk management. On 7 November 2018, a press release
from the prefecture of Mayotte mentioned that the IGN measured a shift of the island eastward
as well as a *"slight downward shift"*. The risk implications were not specified but it was the first
time the volcanological component was officially mentioned, 6 months after the hypothesis
circulated among experts and in the press. The infrasound signal of November 11, 2018, which
occurrence supported the volcanic hypothesis, gave rise to intense discussions among the
international scientific community (Lacassin et al., 2020). It was mentioned by the BRGM in a
news item summarizing current knowledge on the understanding of the ongoing activity published
on its web site on 17 December 2018 (BRGM, 2018b) .
From January 2019, the frequency of BRGM bulletins continued to decrease to reach a
frequency of one bulletin every 20-30 days.

● **Phase B: from February 2019 to February 2020**
On 8 February 2019, following the initiative of the STTM group of Mayotte, 140 inhabitants
of Mayotte signed an open letter addressed to the prefect of Mayotte, the local administration, the
BRGM and the local media. Pressing them for more information (Picard, 2019, on change.org),
they wrote: *"You are not unaware that, for almost 9 months, a large majority of "your" population*
*has been living in anxiety, incomprehension ... Even anguish! The most "basic" questions in terms*
*of security of people, conduct to hold and even projection in the near future ... Are found without*
*any answer! You are certainly convinced that you are doing the maximum so that the panic does*
*not reach your "constituents"? BUT this is not the reality on the ground."* Expectations were
particularly high toward scientists, who were expected to provide explanations and guidance with
respect to risk scenarios. But, in the absence of offshore observations, the scientific advances
were still poor.
February 2019 was an important tipping point, however, as the scientific community finally
received the funding to work in the area. On 22 February 2019, CNRS issued a press release
with the laureates of the Tellus-Mayotte call for tenders (CNRS, 2019). With the launch of the
Tellus Mayotte program, communication opened up to new scientific actors. IPGP and EOST
announced their involvement in the up-coming missions on their website. BRGM scientists
published the first public catalog of the seismic data collected since the beginning of the crisis
(Bertil et al., 2018; Lemoine et al., 2019).

BRGM continued to publish a monthly bulletin dedicated to the monitoring of the seismicity
but communication from the prefecture of Mayotte became more episodic. It focused on relaying
BRGM's situation points (with the list of events - among which the felt ones - in the past months)
and on announcing the arrival of Tellus Mayotte scientific campaigns. The volcanic hypothesis
was eventually put forward in the official communication. The press release of 3 April 2019
mentioned a *"scientific volcanological mission"* aiming at *"consolidating knowledge of the tectonic*
*and volcanic history of Mayotte and at highlighting the tectonic structures of the island by means*
*of dating of magmatic rocks, or analyses of the composition of soil gases"*.

One year after the beginning of the seismic crisis, it was time to take stock of the situation.
In a press release published on 10 May 2019, the Préfecture of Mayotte reviewed the actions
undertaken, both from a scientific and risk management point of view, during the past year, and
concluded that *"the latest data collected by the experts and the modeling of the phenomenon*
*suggested a volcanic origin, possibly linked to a large-scale underwater eruption, or even to an*
*origin combining both tectonic and volcanic phenomena"*. When the scientists of the MAYOBS
campaign arrived at the dock on 16 May 2019, they were accompanied with an interministerial
press release (e.g., *ministère de la Transition écologique et solidaire, ministère de l'Enseignement*
*supérieur, de la recherche et de l'innovation, ministère des Outre-Mer, ministère de l'Intérieur,*
2019) announcing the discovery of a newborn volcano at the origin of the abnormal seismicity
endured by the Mahorais for the past year. The government, through the voice of four of its
ministries, committed to reinforce monitoring and prevention measures[12]. IPGP relayed the press
release on its web site on the very same day (IPGP, 2019a), IFREMER, EOST and BRGM
followed soon after. The announcement was relayed on Twitter, with a spectacular picture of the
underwater volcanic edifice and the rising plume above it (Lacassin, 2019), which raised the
interest of international scientists and of media such as National Geographic, Science, or the BBC
(BBC - Science in Action, 2019; Pease, 2019; Wei-Haas, 2019). The prefecture and vice-rectorate
of Mayotte launched a competition among primary and secondary schools to name the new-born
volcano[13].

There were similar surges of communication after the return of the next marine campaigns
MAYOBS 2 to 4 in June and July 2019, but much less communication afterwards[14]. The effort of
communication resumed again in May 2020 after the MAYOBS13 campaign.

From the discovery of the underwater volcanic activity, the prefecture of Mayotte and the
BRGM were no longer the only two central actors regarding public information. On 28 May, 2019,

---

[12] The press release indicates that the government has defined the following action plan: 1) Complete as soon as possible the monitoring system and install the scientific devices that are necessary to continuously monitor the phenomenon; 2) Complete, through appropriate missions, the scientific knowledge; 3) Immediately update the knowledge of the risks presented by this phenomenon and the potential impacts for the territory of Mayotte; 4) Strengthen without delay the planning and preparation for crisis management ; 5) Regularly inform the population, in conjunction with local elected officials.

[13] The name chosen for the new volcanic edifice was finally made public in December 2021. It did not match the names originally proposed by the children. It is not possible to explain the reasons for this in this paper, as it would require extending our study period. However, it can be noted that the entire process was not consistent with the need to engage people more actively in the recognition of this new source of hazard.

[14] Reports and press releases following MAYOBS campaigns are listed on this dedicated IPGP web page: https://www.ipgp.fr/fr/revosima/rapports-communiques-de-presse-missions-mayobs

BRGM published its latest seismic bulletin on its own and the prefecture of Mayotte published its
latest press release only dedicated to the seismic crisis. Monitoring falled in the hand of the newly
born REVOSIMA. Communication was then discussed at a more centralised level by the
DIRMOM who reported directly to the cabinet of the Prime Minister. The prefecture worked closely
with the DIRMOM to elaborate new communicational tools such as information leaflets. Early
August, the Prefect organized a press conference during which scientists presented the results
of the last campaigns to elected officials and local dignitaries.
The creation of the REVOSIMA was eventually announced one year and four months after
the start of the seismic "crisis" in the end of August 2019, during a visit from the minister of the
Overseas (*Ministre des Outre-mer*) (*Journal de Mayotte*, 27 August 2019). The first web news
concerning the creation of REVOSIMA was published on the IPGP website (IPGP, 2019b).
Entitled *"Volcanological and Seismological Monitoring Network of Mayotte"*, it presented the
mandate of the IPGP and its partners in monitoring the seismic-volcanic crisis in Mayotte.
REVOSIMA issued its first scientific bulletins at the end of August 2019. Several bulletins were
issued approximately at the same time ( (one bulletin for July and two for August 2019) creating
an apparent surge of communication on Figure 4. From then on, two scientific monitoring bulletins
were published every month (it wasreduced to one per month in March 2020)[15].

A scientific conference was organized at IPGP in Paris on 15 October 2019. It aimed to
present scientific advances, and to discuss the challenges of its future monitoring. It was followed
by a public conference and a question-and-answer session in the presence of state
representatives and of the media. It was covered by national media, interested by the
unprecedented nature of the activity (e.g., Vey, 2019), and the local press, proud to see a local
scientist invited (Perzo, 2019b). In October 2019, the Préfecture set up a *"stakeholder
committee"*[16] aimed at bringing together *"all the notables, heads of department, politicians, around
a table"* and to whom scientists would be expected to present, about every six months, *"the
assessment of the crisis and the scientific findings"* (anonymous, interview in May 2020). In
November 2019, the prefecture organised public meetings in several municipalities of Mayotte
but with a sparse audience (a few tens of people, anonymous, interview in May 2020).

In December 2019, the American Geophysical Union fall meeting hosted a special session
dedicated to the Mayotte new volcano discovery where the scientific results from the first
MAYOBS campaigns were presented (e.g., Deplus et al., 2019; Feuillet et al., 2019; Jacques et
al., 2019; Saurel et al., 2019). From our interviews, we understood that some tensions emerged
between the authorities and the scientists about one of the communications (Poulain et al., 2019),
which mentioned a delay of a few minutes between a triggering event due to the volcanic activity
and the arrival of a tsunami on land. The authorities did not want such information to be
communicated without having thought beforehand about the protection measures to be put in
place. The decision was taken to not show the poster (interview in June 2020). At the end of 2019,
EOST also announced the arrival of the second mission of the BCSF-RéNaSS macro-seismic

---

[15] All REVOSIMA bulletins and reports are listed and accessible from the following IPGP web page:
https://www.ipgp.fr/fr/revosima/actualites-reseau
[16] According to our interviewees, this committee has not been very active since its creation. One or two meetings were
organized

intervention group in Mayotte. The continuation of REVOSIMA decided at the December 2019
interministerial meeting was not really announced, at least publicly.
In January 2020, a team of French and German researchers, not members of REVOSIMA,
published in *Nature Geoscience* the first academic paper analysing the evolution in time of the
seismicity and its relation with the ongoing volcanic activity (Cesca et al., 2020). This paper,
mostly based on seismic data acquired by worldwide seismic networks, mentioned the discovery
of the new volcanic edifice before its publication by the scientists directly involved in the survey
campaigns and the close monitoring of the activity. The CNRS and the University of Toulouse,
which hosted the second author of this paper, published a press release in French (CNRS &
Université de Toulouse III, 2020) bearing a sketch section of the proposed magmatic plumbing
system, which was commented by the STTM group: "*So much questions !!! In particular on the*
*position of the magma chamber […] One or Two? 1 or 2 chambers? The island is moving east,*
*towards the supposed chamber near the volcano??? And there's another one just below under*
*the doormat on our front door"*, *"Silly question, but does that portend a big disaster for us?"*
(excerpts from STTM Facebook group, 8 Jan 2020)
In January, EOST also announced the results of the GIM mission and of a pickathon
organized by the REVOSIMA to get help in relocating earthquakes. In February, the BRGM and
the prefecture of Mayotte announced the future launch of seismic-refraction and magnetotelluric
surveys (MAY-MT and REFMAROE).
● **Phase C: From March 2020 to April 2021**
From the beginning of 2020, with the perpetuation of REVOSIMA, the number of actors
communicating diminished. REVOSIMA refocused the communication effort. From March 2020,
the frequency of its scientific bulletins became monthly and automatic bulletins were released
every day online. The monthly bulletins, consisting of about ten to twenty pages, were particularly
appreciated by the scientific community because they contained details on scientific hypotheses,
instruments, methods and results as well as the related uncertainties. Despite a first summary
page aimed at popularizing the contents of the bulletin, they remained nevertheless difficult for
the lay public to access as it was testified of by discussions within the STTM group: *"Gee.... a*
*REVOSIMA bulletin of 21 pages, we didn't expect so much.....I don't understand everything, so I*
*count on THE scientists to tell me if there is something new..."*, and in response, *"Sorry but I can't*
*stand these bulletins anymore! I force myself to read them ? Why : 89 % of repetitions and*
*reminders of the facts ... I haven't read this one yet (the 25th) ! I think that the objective is reached*
*! To make the "average" readers like us run away ! Impossible a short, sharp and clear bulletin*
*??? Saying : "since the last time…"* (excerpts from STTM Facebook group, 5 Jan 2021) and again,
"Silly question, but does it mean a big disaster for us? I have no knowledge on this subject..."
(excerpt from STTM Facebook group, 8 Jan 2021) Shorter exceptional bulletins were issued in
case of felt earthquakes. REVOSIMA monthly and daily bulletins and exceptional press releases
(in case of felt earthquake) were the main supports for information until the end of our period of
study. They were made accessible to the public on a dedicated facebook feed and were regularly
commented on, in the STTM facebook group as well as in the local press. The prefecture
continued to inform the population about new scientific campaigns.
The COVID 19 pandemics, the related lockdowns and travel restrictions complicated the
scientific survey of the crisis. A part of it had to be remotely managed, including the MayOBS13-
2 bathymetric survey in May 2020, operated by a commercial survey vessel while the scientific
team worked on it from their homes. The objectives of these missions were announced by a press
release from the Préfecture of Mayotte (2 May 2020) relayed on the websites of REVOSIMA
partner institutions (IPGP, IFREMER, BRGM). The information was backed up by a governmental
press release (6 May 2020) which recalled *"the state's permanent commitment to protecting the*
*population of Mayotte"* and stated that, as such, *REVOSIMA "[continued] to carry out its land and*
*sea monitoring missions, including in the current health context, with all due precautions"*. Two
information leaflets were also issued that described the release and recovery of OBS (MAYOBS
13-1) and the acquisition of underwater acoustic data (MAYOBS 13-2). While surprisingly, no
press release followed the MayOBS 5 to 12 missions. REVOSIMA issued in May 2020 a detailed
report about MayOBS13 results (REVOSIMA, 2020), which was relayed on the websites of
partner institutions (IPGP, BRGM, IFREMER) on 4 June 2020. The same day, the government
published a press release summarizing the main scientific results and thanking all the staff for
their commitment in these missions.
Two more scientific papers were published in June 2020, one on the volcanological and
seismotectonic context of the seismo-volcanic crisis (Famin et al., 2020), the other one, led by
BRGM scientists, analysed the seismic and GNSS data from the first year (2018-2019) of the
seismo-volcanic episode (Lemoine et al., 2020). A preprint preliminary version of the latter was
publicly available in February 2019 (Lemoine et al., 2019).
The following months were marked by more scattered communications from the
REVOSIMA partner institutions (in addition to the monthly REVOSIMA bulletin), aiming to
summarize the knowledge acquired since the beginning of the crisis (e.g. "two years of seismic
crisis and the birth of an underwater volcano in Mayotte", August 25th, Paquet, 2020). There was
a new surge of communication in October 2020 with the preparation of the MAYOBS-15
campaign. IPGP presented the campaign's objectives on its website on 13 October, 2020 and
published a preliminary assessment of the mission on 29 October (IPGP, 2020). The prefecture
of Mayotte issued a press release presenting MAYOBS-15 results on 28 October. Some of the
scientists of the campaign remained in Mayotte to participate in the "volcano week". Organized
by the prefecture of Mayotte, in close collaboration with the DIRMOM and REVOSIMA, this
"volcano week" aimed to raise awareness of the volcano among the inhabitants of Mayotte. Local
personalities and scientists took turns talking about the ongoing telluric crisis. The scientists
presented their understanding of the ongoing volcanic activity without dwelling on the possible
scenarios. Only the tsunami risk was presented in some detail. Alternative scenarios were shared
to the public recalling that a working group was already working to identify possible evacuation
routes and that a program had been launched to work on a network of sirens and, in the longer
term, a mass alert system by telephone operators. But the information shared during that week
remained quite light on the overall topic of risks and the reactions posted live on the facebook
feed of the prefecture during the presentations were pretty skeptical. The tsunami risk was
commented in the local press as being eventually "quite limited" (Journal de Mayotte, 2
November, YD, 2020). Two presentations by scientists from REVOSIMA were also organized by
the education authority for high school students and 160 science teachers in Mayotte. During the
same week, the prefect of Mayotte inaugurated the first tsunami warning siren in Dembeni and
scientists symbolically handed over volcanic rocks to the Museum of Mayotte. The government
issued a press release on 17 November 2020 that reviewed the results of the MAYOBS-15
campaign and the outputs of the "Volcano Week."

In January 2021, IPGP announced to be the laureate of a major instrumentation project in
Mayotte (Programme Investissement d'Avenir 3, MARMOR project). Led by IFREMER, the
project brings together the core partners of REVOSIMA and prefigures a restructuring of the
governance of research and observation in the region. This change in governance will be all the
more important in the months to come as the DIRMOM's mission ended at the beginning of May
2021, leaving room for a reorganisation within the state services themselves. This reorganisation
is underway at the time of writing and is therefore beyond the scope of this paper. However, it is
interesting to note that our study period, which covers the first three years of the crisis,
corresponds to the first major stage of volcanic risk management in Mayotte.

In March 2021, the researchers involved in the first MAYOBS campaigns and in
REVOSIMA publicly released a preprint of their paper submitted to Nature Geoscience (Feuillet
et al., 2021). This paper was initially submitted to Nature in September 2019, then transferred to
Nature Geoscience in June 2020, but remained confidential until March 2021. It was still under
review after revision at the time of writing. The preprint described the new offshore volcano and
its activity, the evolution of the crisis from the initial deep fracturation processes to the upward
migration of magma across the lithosphere, and discussed the geodynamic context, but did not
discuss future scenarios of evolution and related hazards. Local press summarized its main
results using a lithospheric-scale cross-section from the preprint that illustrated the processes at
work and the location of the seismicity and of magma chambers (YD, 2021). On 15 March 2021,
the online media from the *Cité des Sciences et de l'Industrie* (a science museum in Paris)
published a webdoc summarizing in a popularized way all main results obtained so far on the
Mayotte seismo-volcanic crisis (Minassian, 2021), providing a whole set of new visuals on the
activity. Until then, according to the journalists we interviewed, the coverage of the event was
indeed made very hard by the absence of direct images of the activity. Two main types of images
were used in the official communication as well as in the media: pictures showing oceanographic
vessels or a group of scientists at work and the image showing an underwater plume above the
new volcanic edifice that was made during the first MAYOBS campaigns (Lacassin, 2019 ; Feuillet
et al., 2021).

# 7.    Examining the potential limits of the process of public information with regard to what is known of at-risk populations' information needs

The previous sections aimed at documenting and understanding the organisation and evolution in time of the official response (section 5) and, more specifically, of the process of public information (section 6). We showed that the communication strategy adopted by the local and national authorities in charge of risk and crisis management and by the scientists in charge of monitoring became more structured and more centralised from the summer 2019, with the establishment of a dedicated monitoring body (REVOSIMA) and the support of an interministerial delegation dedicated to major risk reduction in overseas territories (*Délégation interministérielle aux Risques majeurs en Outre-mer,* DIRMOM). We also showed that the number and frequency of public communications had been significant over time, testifying of a constant commitment of these actors to, first, understand and monitor the crisis and, second, communicate their progress publicly. The question that arises then is: how to explain the reported perception of a lack of information among the population? (see sections 3 and 6; Fallou et al., 2020; Devès et al., 2022)? Here we attempt to answer that question by comparing what we learnt about the public information process in Mayotte with what is known, in the literature, of at-risk populations' needs.

The question of at-risk populations' information needs has nourished disaster research for more than 40 years. Excellent summaries of this research exist (e.g. Drabek, 1986; Mileti and Sorensen, 1990; Tierney, Lindell and Perry, 2001). Many studies have focused on how people process and respond to risk communications in emergencies, but the lessons learnt also apply to emergency preparedness efforts - which is the current issue in Mayotte. Lindell et al. (2006) provide a practical summary of what should be known by practitioners in order to design a successful communication strategy. They insist on the fact that people must, first, *receive* information, second, *heed* available information (i.e. pay attention to it) and, third, *comprehend* the information. They broke down information processing into eight stages corresponding to a few typical questions that people ask before making decisions. We summarize these questions below while indicating in brackets the expected outcomes to progress toward protective actions: 1) Is there a real threat that requires my attention? (expected outcome: threat belief), 2) Do I need to take protection action? (protection motivation), 3) What can I do to achieve protection? (decision set), 4) What is the best method of protection? (adaptative plan), 5) Do I need to take protective action now? (threat response), 6) What information do I need to answer my questions? (identified information need), 7) Where and how can I obtain this information (information search plan), 8) Do I need the information now? (decision information). These questions can all be found, in one form or another, on the STTM Facebook publication feed in Mayotte. The people who write on that feed have received information about the activity (they were warned by felt earthquakes and received messages from authorities, the media or peers). However, as Fallou et al. (2020) point out, they complain that the information they receive does not allow them to understand the exact

nature and extent of the threat, and hence to make decisions to prepare or adapt to the associated
risks. Of course, the large uncertainties existing about the activity itself have affected the ability
of authorities and scientists to meet these expectations. But, as we will now see, the public
information strategy that has developed over time has not avoided some well-known pitfalls of
risk communication that would benefit from being corrected in the future.

~~Discussions revolve a lot around scientific knowledge and uncertainties. They are informed~~
~~by publicly available scientific knowledge, in the form of official releases from local authorities,~~
~~scientific reports from institutions involved in monitoring, and more generally anything that can be~~
~~found on the Internet. Fallou et al. (2020) point to the absence of a professional scientist who can~~
~~help the group to translate and contextualize such information. *"The schools for example, which*~~
~~*accommodate some 80,000 students, have been checked by experts (I hope everywhere in*~~
~~*Mayotte) but there has not yet been any feedback to the general public. [...] I would like, for*~~
~~*example, in the general interest, that according to such and such a structure, we could say to*~~
~~*what extent it will resist to such and such a magnitude (including site effects and other local*~~
~~*variables) and also how it will resist to the succession of moderate tremors (in swarm, which is*~~
~~*obviously our case)"* (excerpt from STTM Facebook group, 27 May 2018).~~


Before to go further, it is important to recall that the inhabitants of Mayotte perceive the
existence of offshore volcanic activity only indirectly, mainly through felt earthquakes and,
secondarily, through stories told on social media and in the press or reported, for instance, by
fishermen who observe dead fishes coming up from deep seas. Numerous studies have shown
that experiencing the effects of a hazard increases the attention paid to information about that
hazard (e.g., Sorensen, 2000). From this point of view, it seems reasonable to consider that the
thirst for information of the inhabitants of Mayotte has also evolved during the crisis, in response
to the evolution of the seismicity (Figure 3). The beginning of the crisis was marked by repeated
and strongly felt earthquakes, which goes hand in hand with a strong demand for information
(Fallou et al., 2020). This interest in the topic of earthquakes is further evidenced by a peak in the
number of articles published in the local press at the beginning of the crisis (Devès et al., 2022).
The number of felt earthquakes decreased thereafter and so did interest in earthquake-related
news. This is shown by a significant drop in the number of articles in the local press. Inhabitants
of Mayotte report that, today, the risks associated with the seismic or volcanic activity are barely
mentioned in everyday discussions (anonymous, interview in November 2021). Indeed, people
are exposed to a variety of risks, some of which are more immediate than those associated with
the seismic-volcanic crisis: financial insecurity, energy insecurity, risk of being expelled from the
country, daily struggle for access to water, food, and among the natural hazards, flooding, which
is far more frequent.

### 7.1. The technicalist bias

The public communication is overall characterized by a frequent but minimalist and technicalist discourse. This was particularly true from the beginning of the seismic crisis in May 2018 to the launch of the first scientific campaigns in February/March 2019 (phase A). As expressed on STTM Facebook feed, lists of earthquakes with magnitude and location do not really help people understand the nature or the extent of the threat nor the uncertainties linked to its possible evolution (see section 3, excerpt from STTM Facebook group, 26 May 2018). The frequent use, by scientists as well as by authorities, of specialist terms such as *"risk", "seismic constellation", "magnitude", "intensity",* etc. is another difficulty for those who receive that information. Devès et al. (2022) show that such terms are reproduced in local newspapers without definition or explanation of context.  Among the scientists we interviewed, most argue that "it's not worth worrying people about things that are still hypothetical so [given the uncertainties] we chose to remain very factual" (anonymous, interview in May 2020). But has this *"factual"* communication allowed people to understand *"the big picture"*, i.e. what was happening and what could happen next? We tend to believe that it added confusion by delaying the sharing of robust information. The fact that the Préfecture mentioned the volcanic hypothesis 6 months after the local press undoubtedly contributed to the population's feeling of a lack of information, and also facilitated the emergence of complotism (as documented by Fallou et al., 2020). The technicalist and minimalist tone adopted in official communications was also at odds with the statements that were made by scientists and authorities who insisted on the unprecedented and de facto very uncertain nature of the activity (e.g. the press release of 3 June 2018 stating that "seismic activity remains abnormal and continues").

A final example can be given for illustration here. As reported by Fallou et al. (2020), the fact that some of the felt earthquakes were not reported in scientific bulletins fueled a sense of distrust among the population. Scientists in charge of monitoring took care to publish a note explaining the limitations of the seismic network and the difference with international networks (22 May, BRGM, 2018a). This note was reproduced in part in the local press (e.g. Le Journal de Mayotte, 23 May 2018). But the efforts made to explain instrumental uncertainties were challenged by the technicity of the note, hardly translated by the journalists who copied and pasted whole sections of the text (Devès et al., 2022). Experts' efforts were also challenged by the publication of real-time data, albeit of lower quality, by web applications accessible to all. The prefecture tried to bridge the gap by communicating immediately after earthquakes of magnitude greater than 5 using the data issued by international networks while recalling that *"the estimates of international measurement centers were relayed [...] [waiting for] the BRGM to refine its results"* and that the latter would be *"more accurate because the sensors [were] located in Mayotte and in the area"* (Press release, 5 June 2018). Although this strategy seems legitimate from a scientific point of view, one can wonder if it really helped people to better understand the nature of the existing uncertainties. Indeed, it may seem paradoxical to say that the data is of poor quality when it is *de facto* used in official communication without waiting to be improved.

**7.2. The reassuring bias**
We showed that, beyond the fact that it remained essentially focused on the seismic
hazard, the first phase of communication was marked by the propensity of the various actors of
the risk chain (the authorities, but also the scientists and the local press) to try *"reassuring"* the
population in order to *"avoid panic"*. The local *Journal de Mayotte* reported that *"the mayor of*
*Mamoudzou [was] calling people to calm down and not to give in to any form of panic"* (Journal
of Mayotte, 23 May, Perzo, 2018a). Coming back onto that stage of the crisis, a scientist explains:
"*At the beginning, we talked a lot about the seismic risk to minimize it in the sense that these were*
*only moderate earthquakes, 5.8 was the larger and afterwards we stayed on moderate*
*earthquakes, we communicated quite a lot saying that to have a lot of damage it was necessary*
*to have high enough magnitudes, that it was, maybe, not in the functioning of the system that we*
*knew*" (anonymous scientist, interview in June 2020). After a public press briefing with civil
protection experts and seismologists (Perzo, 2018b), the prefecture posted on Facebook and
Twitter that "there will be no earthquake of a higher magnitude than what we have already known".
And thus, in the local press, one could read that "*Mayotte [was] indeed in a seismic zone, but the*
*tremors [were] not of a nature to worry the scientists*" (Journal de Mayotte, 2 June, Perzo, 2018b).
This attempt to reassure the public by emphasizing the moderate intensity of the threat
had negative side effects when it came to talking about the tsunami threat. The first public
scientific bulletin, published on 16 May 2018, indicated that "in all rigor and given the limited
knowledge in the region, a tremor of magnitude greater than those already observed [could not]
be excluded" and outlined that "these earthquakes [did] not produce damage and, although at
sea, [were] too weak to generate tsunamis" (bulletin of 16 May, BRGM, 2018a). This was taken
up word for word by the officials, and the Minister responsible for the administration of overseas
territories declared the same day that "there [was] no risk of damage on land, nor a tsunami at
sea" (quote from the Ministre des Outre-mer in L'express de Madagascar, 16 May 2018). A few
days later, one could read in national newspapers that: "there [were] no risk of subduction,
therefore there [were] no risk of a tsunami", although "emergency teams [were] ready to be
dispatched from Paris and from Reunion Island where tents and medication [were] stocked", the
journalist outlining that "the watchword [was] to reassure the population." (Le Figaro, 21 May
2018). This press excerpt outlines the paradox of a communication that adopts the tone of
certainty (*"there is no risk"*) and, at the same time, recognizes implicitly the existence of unknowns
(emergency teams are still making ready!). And indeed, a year later, tsunami risk reduction
became one of the priorities of risk management authorities focusing part of the latest
communication efforts[17].
Communication in the context of large uncertainties has proven to be challenging as
contradictions cannot fail to emerge when awareness about the situation becomes more precise.
Devès et al., (2022) point out that news accounts, because of the way they are constructed (by
juxtaposition of remarks made by different actors) tend to highlight these contradictions.
Nevertheless, it remains crucial that authorities and scientists express themselves promptly so as
not to allow space for rumor to gather (see Fallou et al., 2020 on Mayotte's case; Lagadec, 1993

---

[17] The tsunami is one of the first hazards to have given rise to a precise assessment and to the development of concrete preparedness measures (installation of new sirens, definition of evacuation trajectories). Tsunami risk reduction is at the heart of the prevention campaign organized by the DIRMOM in 2021 with videos explaining how to evacuate to higher ground.

or Scanlon, 2007 for general views on the topic). The pitfall here lies in the willingness, often
shared by all the actors (authorities, scientists, and in the case of Mayotte even local journalists
as shown by Devès et al., 2022), to *"reassure"* a supposedly *"panicked"* and *"irrational"*
population[18]. This desire to reassure the population in order to avoid disturbances of public order
is not specific to the case of Mayotte. It has led risk managers' decision making in many other
crises – a famous case is that of Katrina in the United States (Rodriguez, Trainor and Quarantelli,
2006) but examples were also discussed in France (e.g., Borraz, 2019) and about telluric
phenomena such as earthquake sequences (e.g., L'Aquila, see discussion in Cocco et al., 2015;
Jordan, 2013). However, the representations of *"officials [who] must be careful about issuing*
*warnings because of the danger of panic"* and *"victims [who] will be dazed and confused, perhaps*
*in shock, and must be cared for by others"* (Scanlon, 2007: p. 416) have been shown to be
*"inaccurate, biased and often exaggerated"* (Rodriguez *et al.*, 2007: p. 482). They corroborate
certain myths circulating in society, largely deconstructed by the social sciences (Mileti, 1999).
The populations facing extreme situations, rather than becoming confused, passive and irrational,
are on the contrary extremely pragmatic and proactive and tend to react by reinforcing social
control mechanisms to face danger (Quarantelli, 2008; Solnit, 2010).
Sharing experiences, emotions and information on a Facebook publication feed is an
interesting way to collectively manage stressful situations. But, when scientific knowledge is
concerned, the ability to select and comprehend information soon becomes a crucial issue (see
the excerpt from STTM Facebook group, 8 Jan 2021, section 6). Fallou et al. (2020) report that
the members of the STTM Facebook group worked at describing the phenomenon as accurately
as possible (following the group, you could know whenever an earthquake was felt, with which
intensity and what impact from place to place) and at bringing together all the information they
could find (sources were official releases from local authorities, scientific reports from scientific
organisations involved in monitoring, and more generally anything that can be found on the
Internet, see Fallou et al., 2020). They also point to the absence of a professional scientist who
could help the group to translate and contextualize this information. The question arises of the
role to be played here by the scientific community. It is true that, given the uncertainties, some
questions could not be answered but, as suggested by Lindell et al. (2006), one might have
explained earlier what was known and not known, and what could be done to address that lack
of knowledge. As noted by Sharma & Patt (2012), empirical studies tend to show that *"lay people*
*do understand uncertainty and, under conditions of good communication, even understand*
*probabilistic forecasts. Therefore, there may be value in communicating uncertainty from the point*
*of view of improving the credibility of the message."* This is particularly important as many studies
have shown that the experience about the credibility of the message affects the response to
warning in the next future event (Lindell et al., 2006; Sorensen and Sorensen, 2018). The recent
development in research about uncertainty communication can help designing communication
strategies in this respect (see Doyle et al., 2019 for an overview). This requires scientists to adapt
their practices because, as concluded by Doyle et al. (2019), *"scientists must first understand*
*decision-maker needs [and we add here that at-risk populations are not the least of the decision-*

---

[18] Devès et al. (2022) analyse the representation of authorities, scientists and inhabitants in media accounts and show that the place they are ascribed to echoes disaster myths (Quarantelli, 2008). This is well illustrated in the following press excerpt: "Many irrational reactions, faced with which the BRGM explains…" (Le Journal de Mayotte, 23 May 2018)

*makers in case of emergencies], and then concentrate efforts on evaluating and communicating*
*the decision-relevant uncertainties."*

**7.3. The hazard bias and the lack of risk scenarios**

We showed that, from the launch of the first scientific campaigns in February/March 2019
to the creation and perpetuation of the REVOSIMA (phase B), the format and the nature of
communication changed. At first, it was distributed among much more actors and then changed
scale with a resumption of communication by national actors (major scientific institutions, CNRS,
ministries and government through the DIRMOM). But it remained relatively coherent as each of
these actors were referring to the joint Tellus Mayotte work program in their communications. The
discoveries made during the MAYOBS1-2 and MAYOBS 3-4 missions constituted an important
turning point in the content of the information that was shared. From May 2019, communications
no longer focused only on seismic hazard but started drawing a more general explanatory
framework attributing earthquakes to an offshore, and unexpected, volcanic activity. But despite
this important change, the communication remained centered on hazards rather than on risks,
which still does not allow answering the population information needs. Reading the press and the
STTM facebook feed, one realizes that people were excited by the unprecedented scientific
mobilisation around their island and expected to learn a lot from scientists. But after the first
campaigns, given the extent of the discovery that made fear of potentially high associated risks,
the authorities became very cautious about communication. They asked the scientists to refine
their scenarios before sharing openly information about risks with the population (we mentioned
earlier some tensions in AGU). A scientist reports that *"today [a year after the discovery of the*
*volcano] we are starting to talk about all the risks. But we are talking about it with frilosity. But it*
*is not the scientists who talk about it with frilosity, I think that the authorities have locked up this*
*subject a little."* (anonymous, interview in May 2020). Some of the scientists actually share the
frilosity of the risk managers pointing out that *"I prefer to publish, and to get a peer-to-peer*
*validation of my hypotheses, before sharing them publicly [...] I don't want to panic people"*
(anonymous, interview in July 2020). Hence, public information tended to settle for highlighting
the unprecedented nature of volcanic activity and the prowess scientists had to deploy to study it.
Little was said about the possible evolution of the hazard although, as recalled by another
scientist, *"we identified [coarsely] the possible scenarios probably from May-June 2019"*
(anonymous, interview in May 2020). On STTM Facebook Publication feed, the feeling prevailed
that communication did not answer the important questions: *"[…] The state gives up a lot of money*
*and resources... But no respect for the population! No info (the same for 2 years! True!) No*
*listening to people and their requests! No explanation in the villages […] And when they give a*
*conference (scientific or press) it is to repeat the same information over and over!" (*excerpt from
STTM Facebook group, 5 Jan 2021)*.*
So far, i.e. three years after the beginning of the seismic crisis, scenarios have only been
communicated orally, in the form of a listing of potential hazards, indicating that scientists are still
working to refine their assessment of the associated risks. But this strategy is debated among
scientists. Some argue that *"these are still scenarios, so we must be very careful [in*
*communicating] [...] I understand that some scientists are a little confused because a lot of work*
*has been done and not all the information has been passed on to the general public, but I think*
*that the general public does not need to know certain information either, because it is all just*
*hypotheses and then you take a sentence out of context and it's panic. I understand that"*
(anonymous, interview in May 2020). Others respond: *"I think it's better [...] that people are aware*
*that one day there could be a mudslide in their garden or a tsunami than not to know. I know that*
*Mayotte is maybe more complicated because, I don't know, they have other problems but it's not*
*a reason to hide it from [people]..."* (anonymous, interview in June 2020). Between the supporters
of a communication based on certainties and quantitative assessment, which is structurally close
to the strategy adopted by the authorities, and the supporters of a certain level of academic
freedom in communicating hypotheses at work and not just confirmed results, the debate is still
open.
Both strategies have advantages and caveats. Davies et al. (2015) argue that *"quantitative*
*risk assessment and risk management processes"* are *"of value at regional or larger scales by*
*governments and insurance companies"* but do not provide *"a rational basis for reducing the*
*impacts at the local (community) level because in any given locality disaster events occur too*
*infrequently for their future occurrence in a realistic timeframe to be accurately predicted by*
*statistics"*. They suggest, instead, that *"communities, local government officials, civil society*
*organisations and scientists could form teams to co-develop local hazard event and effects*
*scenarios, around which the teams can then develop realistic long-term plans for building local*
*resilience"*. As outlined by earlier studies, as providers of the primary information about the
hazards, scientists are - whether they like it or not - at the heart of the risk reduction process (e.g.
Rodriguez et al., 2017; Donovan, 2021). They cannot wait for the very last quantitative results to
share their knowledge, i.e. their hypothesis, their methods and their results (that can be negative
ones proving that an hypothesis does not hold). They have a moral, when not legal, responsibility
to respond to the demand for information from different audiences (authorities, people likely to be
affected, journalists, etc.) and at all times (times of larger or smaller uncertainties). Jasanoff
(2005) speaks about *"civic epistemology"* as *"the institutionalized practices by which members of*
*a given society test knowledge claims used as a basis for making collective choices"*. Scientists'
role is indeed all the more central as their opinions not only inform, but also legitimize the
decisions taken by the authorities in charge of civil protection and risk management. Of course,
such a posture is not easy to adopt, notably because there is a bounded understanding of the
scientific approach in our societies (e.g., Bromme & Goldman, 2014). During our interviews, we
were said that the comments posted on STTM hurted some scientists. Referring to the criticisms
read on the Facebook of the STTM group, one of them says: *"What they did not understand is*
*that we did not understand what was happening either [...] Because there is no analog [...] We*
*started from an area considered as [inactive]. We find ourselves in an unknown zone to manage*
*a phenomenon without analogue while having to organize missions involving unprecedented*
*means [i.e. large scientific boats that should be booked months in advance] [...] Our role is to*
*make scientific reports [but] I think these have a limited impact [because] there is no one on the*
*ground [who can translate what we do]."* (anonymous, Interview in July 2020). That such
knowledge *"translation"* has to be done by concerned scientists actively engaged in science
communication and in answering people's concerns, or by professional *"knowledge brokers"*
(Hering, 2016), is an open question.
The publication of an article by REVOSIMA researchers on EarthArxiv (Feuillet et al.,
2021) in march 2021 gave rise to mixed feelings in the STTM feed. The fact that the publication

was not associated with a document in French and addressed to the lay public was not much appreciated: *"they are seriously starting to get on my nerves! A choice to address only peers! And damn for a minimum of popularization and "simple" explanations. Afterwards, they are surprised that some and others tell everything, anything! or blame them for their "Height""* (excerpt from STTM Facebook group, 17 March 2021). The intuitive interpretations they made of the article, from the point of view of risks, was rather accurate: *"I learn from this cross-section that the volcano's chimney is 15km from Mamoudzou and not 50, where the underwater volcano is formed. Not reassuring. Moreover, the last activities mentioned are in the main volcano, so very close to us."* (excerpt from STTM Facebook group, 17 March 2021). People have clearly understood that it is not the new volcanic edifice that poses a significant risk to them. They are very concerned about the seismicity located closer to the island, especially since the publication of the cross-sectional diagrams of Cesca et al. (2020) and Feuillet et al. (2021). They ask themselves questions about a future eruption very close, and/or collapse on the outer-reef slope generating tsunamis, which corresponds more or less to the scenarios considered by scientists. To this respect, it seems rather vain not to communicate on scenarios.

### 7.4. The complexity of multiculturalism

To conclude this discussion, it is important to come back to an essential fact about risk reduction in Mayotte in its communication aspect. Lindell et al. (2006) emphasize that for individuals to effectively adapt their response to a risky situation, they must not only receive information, but also consider and understand it. It is clear that individuals comprehend information only if it is provided in a language they understand, at a time and in a format they are accustomed to use. The above discussion shows that even if information is shared publicly, it is not properly formatted to be understood even by the part of the population investing time to dive into the topic. Risk communication in multicultural contexts, and on a small island, poses specific challenges (e.g. Lindell and Perry, 2004 or more recently Bolin, 2018 about race, ethnicity and vulnerability; e.g. Koromowski et al., 2018 on the challenges of risk communication on small islands). The fact that written communication to date has been primarily in French, an official language but one that is far from being well understood by the majority of the population, is a major problem. Efforts have been made to translate some of the communication materials, including the seismic safety guidelines, into Shimaoré in May 2020, but this is far from sufficient. Identifying the various habits of the population with respect to communication (not only language but also practices, who listens to who?) would also be important to adapt both format and contents. As pointed out by the Senator of Mayotte, Thani Mohamed Soilihi, orality plays an important role in Mayotte and written formats would gain to be accompanied orally (radio, animated movies but also neighborhood meetings and informal discussions with prominent members of the various social groups composing Mayotte (associations, muslim religious chiefs, etc.) (interview excerpt in the Report of activity of the DIRMOM, May 2019 - July 2020).

# 8.    Conclusions
As pointed out by Stewart and Lewis (2017), *"scientists' attention to technical accuracy*
*and their emphasis on professional consensus may do little to influence multiple publics whose*
*worries instead root into their sense of place, trust and governance, as well as equity and ethics."*
The work done on the circulation of information from its place of production (the laboratory, the
boat, the field) to different publics (authorities, media, population) during the first three years of
the Mayotte seismo-volcanic crisis supports this observation (also see Devès et al., 2022). As
outlined by many earlier studies, there are cultural differences between scientists, authorities and
at-risk populations (e.g. Newhall, 2017; Haynes et al., 2008 for discussion on volcanic cases). We
can only agree with Newhall (2017) when he writes that *"trying to understand and accept the*
*cultural differences among the various groups [he refers here to scientists and authorities but one*
*can add populations, medias, …], and involving users in the scientific process whenever feasible,*
*are the best ways … to develop this thrust"* which *"is essential if that information is to be accepted*
*and used"*.
The efforts made by the risk chain actors to share information are undeniable, as well as
the knowledge built up over time at the cost of a high level of commitment (from the Prime
Minister's office to ship technicians). This is reflected in a significant volume of publications that
take various forms, from press releases to scientific bulletins, web news or communication events.
But the effort is insufficient insofar as it does not allow to reach *"the last mile"* (e.g., Shah, 2006)
towards the populations. Many factors come into play here, some of which are well known to the
social sciences, and some of which have to do with the complicated relations between
metropolitan France and the French overseas territories.
In terms of communication there are several possible ways to gain efficiency. The first
consists in establishing a real strategy of research and expertise dedicated not only to hazard
monitoring but more broadly to the reduction of risks, the latter being considered in their technical
dimension but also in their human and social aspects. The second is to work on the content and
formats of information sharing. As emphasized by Oreskes (2015) about seismic risk, *"earthquake*
*safety has never been simply a matter of geophysics, but most earthquake scientists, acting qua*
*scientists, have traditionally understood their job to be to study how, when, and why earthquakes*
*happen, and only to a lesser extent (if at all) how to communicate that knowledge to engineers*
*and officials responsible for mitigation, or to the general public [...] But in the contemporary world,*
*the inter-relationship between knowledge and safety is not easily disentangled. Seismology is no*
*longer simply a matter of geophysics, if it ever was. It involves consideration of ethics, values,*
*and monetary and social costs. [The trial of] L'Aquila shows that scientists can no longer ignore*
*the social factors that affect and even control how damaging a particular earthquake may be.*
*Earthquake prediction is a social science."* The reasoning applies to the assessment of other
*"natural"* risks. If scientists' main job is not to communicate, they are nevertheless the only ones
able to appreciate the robustness of the science-based information. As such, they are expected
to take the time to present it in a way that can help risk managers, elected officials, journalists
and the wider population to act effectively. From this point of view, it seems important to work at
clarifying the frontier between the communication of scientific advances on hazard understanding,
and the communication of operational risk management measures. That frontier seems
particularly blurry in the case of Mayotte. The advantage of this clarification would be twofold.
Allowing scientists to explain their hypotheses, results and uncertainties would lead to an
improvement of the population's scientific culture while reinforcing the credibility of the scientific
expertise. The latter is a pillar of any science-based risk governance process, as one may adhere
to decisions made by authorities only if he/she believes their scientific basis to be credible. The
adhesion to the scientific approach is thus a prerequisite to the adhesion to the risk reduction
approach carried out by the other actors of the chain. The third lever is the association of local
personalities, elected officials, local NGOs, to the reflection on the risk scenarios and adaptation
strategies. The international Sendai Framework for Disaster Risk Reduction calls for a more
integrated practice. The signatory countries reckon that, in order to reduce efficiently the risk of
disasters, *"there is a need for the public and private sectors and civil society organisations, as*
*well as academia and scientific and research institutions, to work more closely together and to*
*create opportunities for collaboration […]"* (Sendai framework page 7 - UNISDR, 2015). Following
Ismail-Zadeh et al. (2017), Stewart, Ickert and Lacassin (2018) emphasize that the willingness for
greater integration defines a *"new social contract between hazard scientists and the wider public*
*[...] that encourages the scientific community to endeavour, alongside their existing technical*
*expertise, to '... support action by local communities and authorities; and support the interface*
*between policy and science for decision-making' (Sendai framework page 22 - UNISDR, 2015)"*.
As shown in this paper, this change of expectations creates new challenges for scientists, notably
on the issue of communication. We hope that this work will contribute to open new leads for
transdisciplinary research drawing on geosciences, social sciences and humanities that can
improve the effectiveness of the science-society nexus for disaster risk reduction.

# Data availability

EMSC data on the felt seismicity are available from https://doi.org/10.5281/zenodo.4734032.
Instrumental seismicity plotted on Figure 1 is from Lemoine et al. (2020) dataset, and from
REVOSIMA catalog (not yet available for distribution, these data will be included in Saurel et al.,
2021). A table listing all the written documents issued by the scientific and state institutions
involved in monitoring and risk management is provided in supplementary information. The press
releases from the prefecture de Mayotte and French ministries that we refer to in the text are
given in full in supplementary dataset. French version of STTM post excerpts are also provided.
Full verbatim of interviews from which we extracted cited excerpts are not public for confidentiality.
All other data used in this paper are available from cited references.

# Author contribution

MHD was responsible for the conceptualization of the study, project administration, methodology
and writing the original draft of the paper. MHD and RL undertook the revision and editing of the
final paper in concert with all co-authors. MHD and GR were responsible for data curation and
investigation. RL curated the STTM Facebook threads and selected relevant excerpts. MHD and
GR conducted and transcribed the interviews. MHD, RL and GR undertook the formal analysis.
MHD and RL carried out the validation. HP, RL and MHD were responsible for the figures.

# Acknowledgements

This work benefited from the data of felt seismicity collected by the EMSC (https://doi.org/10.5281/zenodo.4734032) whose activities is supported by the *Fondation SCOR pour la Science*. The authors would also like to thank Rémy Bossu, Laure Fallou and Matthieu Landès from EMSC who provided very useful comments at different stages of the work. People interviewed during these three years are also to be thanked for their invaluable contribution to our thinking process. The authors also want to thank Emmelyne Mitard, communication officer in IPGP, scientists and officials from the DIRMOM, the prefecture of Mayotte and the REVOSIMA for their help in setting up exhaustive records on the communication elements put in place by the scientists and the authorities during the course of the crisis. Many thanks also to Aline Peltier who chairs the scientific committee of REVOSIMA and kindly agreed to read and comment on the paper before its submission. And finally, thanks to Louise Le Vagueresse and Hugo Pierrot, whose work helped us to check the completeness of our communication database.

This research has been supported by the IdEx Université de Paris, Centre des Politiques de la Terre, ANR-18-IDEX-0001.

# Competing interests

The authors declare that they have no conflict of interest.

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
