# Peer review of "Risk communication during seismo volcanic crises: the example of Mayotte,"

_Natural Hazards and Earth System Sciences, 2021_

## Author Comment (AC1)

[revised manuscript text omitted]
 a supprimé: s a supprimé: is a supprimé: is a supprimé: which was not a supprimé: to be particularly active seismically a supprimé: In this context, at the beginning of the crisis, a supprimé: is a supprimé: It should however be noted that, for BRGM, as for the other scientific organizations involved in monitoring before the creation of REVOSIMA, real-time data processing is organized thanks to scientists' voluntary commitment. …

a supprimé: is a supprimé: are a supprimé: are a supprimé: does a supprimé: leads a supprimé: other a supprimé: to become involved a supprimé: d a supprimé: .of risk and crisis management experts. The arrival of the interministerial mission composed of civil protection experts and seismologists in early June 2018 is an opportunity to take a step back from the situation a supprimé: is a mis en forme : Couleur de police : Texte 1

a mis en forme : Couleur de police : Texte 1

a mis en forme : Couleur de police : Texte 1

a supprimé: are a supprimé: as its intensity decreases

Commenté [2]: A note was added here in response to an external review received by email.

a mis en forme : Police :9 pt, Couleur de police : Automatique a mis en forme : Police :9 pt, Couleur de police : Automatique

[revised manuscript text omitted]

a supprimé: i
a supprimé: ng
a supprimé: ying
a supprimé: carrying
a supprimé: is
a supprimé: o
a supprimé: is
a supprimé: The pPréfecture and vice-rectorate of Mayotte launched a competition among primary an ... [2]
a supprimé: will
a supprimé: that will allowed refining
a supprimé: s
a supprimé: toof crisis monitor
a supprimé: ing
a supprimé: s
a supprimé: are
a supprimé: is
a supprimé: is
a supprimé: operates
a supprimé: is
a supprimé: expected to set up specific scientific a ... [3]
a supprimé:
a supprimé: is
a supprimé: geo
a supprimé: of REVOSIMA
a supprimé: of the seismicity
a supprimé: perpetuation
a supprimé: of Mayotte
a supprimé: s
a supprimé: development
a supprimé: new
a supprimé: 8
a supprimé: s
a supprimé: is
a supprimé: o
a supprimé: is
a supprimé: ve
a supprimé: is
a supprimé: mission
a supprimé: is

[revised manuscript text omitted]

Police :9 pt, Couleur de police : Automatique

| Page 25 : [8] a supprimé | Maud Devès | 02/01/2022 13:37:00 |
|---|---|---|

| Page 25 : [8] a supprimé | Maud Devès | 02/01/2022 13:37:00 |
|---|---|---|

| Page 25 : [8] a supprimé | Maud Devès | 02/01/2022 13:37:00 |
|---|---|---|

| Page 25 : [8] a supprimé | Maud Devès | 02/01/2022 13:37:00 |
|---|---|---|

| Page 25 : [8] a supprimé | Maud Devès | 02/01/2022 13:37:00 |
|---|---|---|

| Page 25 : [8] a supprimé | Maud Devès | 02/01/2022 13:37:00 |
|---|---|---|

| Page 25 : [8] a supprimé | Maud Devès | 02/01/2022 13:37:00 |
|---|---|---|

| Page 25 : [8] a supprimé | Maud Devès | 02/01/2022 13:37:00 |
|---|---|---|

| Page 25 : [9] a supprimé | Maud Devès | 02/01/2022 13:38:00 |
|---|---|---|

| Page 25 : [9] a supprimé | Maud Devès | 02/01/2022 13:38:00 |
|---|---|---|

| Page 25 : [9] a supprimé | Maud Devès | 02/01/2022 13:38:00 |
|---|---|---|

| Page 25 : [9] a supprimé | Maud Devès | 02/01/2022 13:38:00 |
|---|---|---|

| Page 25 : [9] a supprimé | Maud Devès | 02/01/2022 13:38:00 |
|---|---|---|

| Page 25 : [9] a supprimé | Maud Devès | 02/01/2022 13:38:00 |
|---|---|---|

| | | |
|---|---|---|
| **Page 25 : [9] a supprimé** | **Maud Devès** | **02/01/2022 13:38:00** |
| **Page 25 : [9] a supprimé** | **Maud Devès** | **02/01/2022 13:38:00** |
| **Page 25 : [9] a supprimé** | **Maud Devès** | **02/01/2022 13:38:00** |
| **Page 25 : [9] a supprimé** | **Maud Devès** | **02/01/2022 13:38:00** |
| **Page 25 : [9] a supprimé** | **Maud Devès** | **02/01/2022 13:38:00** |
| **Page 25 : [9] a supprimé** | **Maud Devès** | **02/01/2022 13:38:00** |
| **Page 25 : [9] a supprimé** | **Maud Devès** | **02/01/2022 13:38:00** |
| **Page 25 : [9] a supprimé** | **Maud Devès** | **02/01/2022 13:38:00** |
| **Page 25 : [10] a supprimé** | **Maud Devès** | **02/01/2022 13:39:00** |
| **Page 25 : [10] a supprimé** | **Maud Devès** | **02/01/2022 13:39:00** |
| **Page 25 : [10] a supprimé** | **Maud Devès** | **02/01/2022 13:39:00** |
| **Page 25 : [10] a supprimé** | **Maud Devès** | **02/01/2022 13:39:00** |
| **Page 25 : [10] a supprimé** | **Maud Devès** | **02/01/2022 13:39:00** |
| **Page 25 : [10] a supprimé** | **Maud Devès** | **02/01/2022 13:39:00** |
| **Page 25 : [10] a supprimé** | **Maud Devès** | **02/01/2022 13:39:00** |
| **Page 25 : [10] a supprimé** | **Maud Devès** | **02/01/2022 13:39:00** |
| **Page 25 : [10] a supprimé** | **Maud Devès** | **02/01/2022 13:39:00** |
| **Page 25 : [10] a supprimé** | **Maud Devès** | **02/01/2022 13:39:00** |
| **Page 25 : [10] a supprimé** | **Maud Devès** | **02/01/2022 13:39:00** |

| Page 25 : [10] a supprimé | Maud Devès | 02/01/2022 13:39:00 |
| Page 25 : [11] a supprimé | Maud Devès | 02/01/2022 14:03:00 |
| Page 25 : [11] a supprimé | Maud Devès | 02/01/2022 14:03:00 |
| Page 25 : [11] a supprimé | Maud Devès | 02/01/2022 14:03:00 |
| Page 25 : [11] a supprimé | Maud Devès | 02/01/2022 14:03:00 |
| Page 25 : [11] a supprimé | Maud Devès | 02/01/2022 14:03:00 |
| Page 25 : [11] a supprimé | Maud Devès | 02/01/2022 14:03:00 |
| Page 25 : [11] a supprimé | Maud Devès | 02/01/2022 14:03:00 |

---

## Author Response (AR1)

Dear Editor,
Dear reviewers,

Please find enclosed the revised version of the manuscript. We found the reviewers' comments particularly useful and implemented the majority of the requested changes. These are discussed point by point below. You can consult the details of the changes in the document titled *revisedmanuscript_withtrackchanges* with the functionality "track changes". The final version of the revised manuscript, titled *revisedmanuscript_final*, has been compiled by accepting all changes.

Note that we also received comments by email from scientists involved in the monitoring of the crisis and from the service in charge of risk communication in the French minister of Environment. We introduced some changes in response to discussions with them (they are notified in comments in the manuscript with track changes). These are minor changes that do not alter the original message of the paper and that are not in contradiction with the demands of the reviewers.

We believe that the suggested changes have significantly improved the paper and we hope you will find it ready for publication, but we remain at your disposal for any further improvements you might find necessary. Thanks again for your help in dealing with this manuscript.

Sincerely,

Maud Devès and co-authors

**Point-by-point response to the reviews**

**General comments from both referees**
Both referees seem to agree on the fact that the paper makes an important contribution and deserves to be published in NHESS. But they also agree on the fact that it needs some revisions. They express three main requests: 1) more developments on the review of literature on volcanic risk communication, 2) a more robust presentation of the methodology and 3) a clarification of the criteria used to differentiate the crisis management vs communication phases. Anna Hicks also wonders about mixing the results and discussions parts of the paper to avoid redundancy. In the following, we begin by responding to these four main requests. We then describe how we responded to "specific comments" and "technical corrections". Before diving into all that, we want to warmly thank them both for providing a lot of very helpful advice.

1. **more developments on the review of literature on volcanic risk communication**

We agree that the introduction deserved to be reworked to clarify the place of the proposed work in the existing literature about volcanic risk communication but also the broader field of disaster risk reduction. We thought it would be clearer to create three distinct sections (sections 1, 2, 3 in our revised manuscript):
1) the introduction is now dedicated to the literature review and situate our work with more clarity (at least we hope so), it also explains the structure of the paper,
2) the second section is dedicated to Mayotte's geological setting and reviewing what is known today of the ongoing activity - it was also a demand of the reviewers to expand on that aspect, hence we did it while trying to keep a reasonable length to the paper,
3) the third section builds on the previous version of the introduction to present elements of the social and political context that contributed to transform a seismo-volcanic phenomenon with, so far, relatively minor consequence, into what had been called a "communication crisis" by risk managers (interviews).
In the end, the first 7 to 8 pages of the initial version of the paper were completely rewritten, answering most of the "specific comments" and "technical corrections" made on these pages (see below our response to the ones left). We hope that these substantial changes will answer the reviewers' concerns. Please do not hesitate to suggest more references if you still find it necessary.
Note that we also reworked the discussion section (called section 7 in our revised manuscript) and completed it with additional references to strengthen the arguments the reviewers commented on.

2. **more robust presentation of the methodology**

We agree that the methodology section deserved to be reworked to describe more precisely the empirical data and the methods used to conduct this research.

We now start by situating our work within its broader context. The present research is part of a research project entitled MAY'VOLCANO dedicated to the study of the circulation of knowledge between scientists, risk and crisis management actors, the media and the

population of Mayotte during the seismo-volcanic crisis. This paper aims at providing a first analytical view of the public information process, and of its potential limitations. A second paper devoted to the study of media narratives using discourse analysis methods is currently being revised (Devès et al., under review). More specific analyses, conducted in collaboration with language and communication specialists, will follow. The use of those methodologies are however out of the scope of the present contribution.

We restructured the methodology chapter (now section 4 in our revised manuscript) into 3 subsections that mimic the organization of the paper: 4.1) documenting and understanding the organisation of the "official response" and its evolution over time, 4.2) documenting and understanding the organisation of the public information process and its evolution over time, and 4.3) examining the public information process in light of what is known about the information needs of the population at risk. In each of these sub-sections, we explain our objectives and how we tried to achieve them. In particular, we provide more details about the way we dealt with semi-structured interviews and with participant observation (note that we corrected the number of semi-structured interviews because we undertook a few more since the submission of the paper). We also explain in more detail which data were collected to assess the volume, timing and content of public information and we explain how these were treated (the use of the word coded was a mistake due to a bad translation). To respond briefly to the reviewers here, each item was downloaded, given a specific ID, stored, and then read independently by 2 to 3 researchers who completed a shared table with information about format and content (disagreements were discussed and solved collectively). We took note of the ID, the date of publication, the URL (when existing), the publishing authors/institutions, the title, the public it aimed to, the number of words, the presence or absence of illustrations and the nature of these illustrations (scientific or not). We also took note of the main topics covered by the text and of the actors that were mentioned. This dataset was used to quantify the volume and timing of public information. It also served as a guideline for a qualitative analysis of content. As stated earlier, we did not wish to use, for this paper at least, quantitative discourse analysis methods because its aim was broader. The chosen method places emphasis on the meaning rather than the quantification of the materials, which we believe is appropriate to the present research.

3. **clarification of the criteria used to differentiate the crisis management vs communication phases**

This is a very good remark. We tried to be clearer in the new introduction and methodology sections. As it is now explained in the introduction, as no emergency planning or monitoring had ever been done in the department of Mayotte with respect to volcanic issues before May 2018, the framing of the official response has evolved significantly over time, and documenting this evolution had been a significant part of our work. Section 5 of our revised manuscript (corresponding to former section 3) provides an informed narrative of the successive stages of organisation of monitoring and risk management: four main phases (1, 2, 3, 4) are distinguished and they are presented in a chronological way. Because public information strategies have not always evolved in tandem with monitoring and risk management frameworks, communication issues are discussed separately in section 6. Analysis of the volume, timing and content of the written documents used by authorities and

scientists to share information with the public led us to distinguish three main phases of communication (A, B, C). Those are not coincident with the former 4 phases. Of course, we agree that communication is an intrinsic and necessary part of risk and crisis management (it is the whole point of this work!) but we observe that it has not always been thought of as such by the actors themselves. Moreover, as explained in our previous response about methodology, we thought it would be easier to adopt a structure that mimics the various phases of the work done and, documenting and understanding the organisation of the "official response" and its evolution over time, does not involve exactly the same methods and datasets as "documenting and understanding the organisation of the process of public information and its evolution over time". Differences should be better explained now in the new methodological section.

**4. Mixing the results and discussions parts of the paper to avoid redundancy**

We got the point about the organisation of the last part of the manuscript and we debated about it between us. It was finally decided to keep the structure of a typical "earth sciences" paper because the majority of us thought it would be clearer for the readership of NHESS. However, as emphasized earlier, the new introduction and methodology sections should help make the structure clearer to the reader as the logic of it is now explained more explicitly: first, documenting and understanding the organisation of the "official response" and its evolution over time, second, documenting and understanding the organisation of the public information process and its evolution over time, and third, examining the public information process in light of what is known about the information needs of the population at risk. We also moved bits and pieces of the text from the results (now section 6) to the discussion part (now section 7) in order to avoid repetitions. We think it is clearer now.

**Referee 1: Anna Hicks' specific comments**

16-29 This research stems from such an interesting event and at the moment the abstract falls a bit flat. Its weakness in the first half is partly due to grammatical errors which could be easily rectified. The second half is slightly better, and promises the reader that the results and recommendations from this case study analysis will be situated within the wider geo-risk communication literature/case studies, which unfortunately doesn't quite come out strongly in the analysis later on (see later comments). In line 20, I would suggest revising the use of 'publications', given that the audience in this case will largely be academic and publications means something specific. I would argue that the communication strategy was not 'put in place' for the first three years of the crisis – whatever 'strategy' was designed at the outset of the crisis was clearly revised and reiterated later on. I would also urge the authors to be careful of, and revise any use of conjecture, in the abstract and elsewhere in the text e.g. "we notably stress". Do you mean, "we present evidence for the importance of?"

+

31 The first paragraph is very unclear and, similarly to the abstract, could present the problem and the research much better. You have a great story to tell, so open strongly! As a start, please change the phrase a 'very active seismic crisis' - this is not quite right. It would also help to refer the reader more quickly to Figure 1 (i.e. in the first sentence or two), particularly if they are unfamiliar with the geography/geology of Mayotte.

→ The abstract was rewritten and the introduction too. See our response above, page 2, section 1.

39 I would like to read a little more about the geological setting. Could the sentence beginning Mayotte become a separate paragraph and have slightly more detail added to the hypotheses for volcanism in the region? There's more information provided about the monitoring of a nearby volcano than there is about the tectonic/volcanic setting! This is an understudied region and I'm sure the readers would like to know a little more.

→ We created a new introductive section (section 2 in our revised manuscript) dedicated to Mayotte's geological setting and to what is known today of the ongoing activity. Also see our response above, page 2, section 1.

69 Can a reference be added to the note about the population's distrust in the authorities? How do the authors know this?

+

109 Distrust in state services. Similarly to my earlier point, this is interesting and warrants a reference if you have one.

→ We reworked the corresponding paragraph to add a bit more information about the social context of Mayotte. Distrust in state services is quite common in many of the French overseas departments, relationships with metropolitan France being complicated by the colonial past. This is something that is often briefly mentioned by authors working in these areas. Mori (2021) mentions it in her recent paper about the seismic crisis in Mayotte. It is also mentioned by Komorowski et al. (2018) in "Challenges of Volcanic Crises on Small Islands States". In Mayotte, the context is however a bit different from the other French overseas (see Regnault, 2011). Anthropologists agree that there is no real integration between the traditional culture of the villages and the more westernized culture of large cities (Lambek, 2018). According to Regnault (2011), "three quarters of the *Mahorais* - rural or, at least, still very attached to their village - live a culture other than the "westernized" culture of the cities" (*trad. by the authors*). The relationship with state authorities is complicated by the island's colonial past, but also by a sense of disappointment among the population, who expected more rapid changes to bring the island up to French standards after departmentalization (Roinsard, 2019). Since 2011, Mayotte has been regularly shaken by social crises. The most recent one, which brought the economy to a standstill for two months in the spring of 2018, was just ending as the first earthquakes began (Roinsard, 2019; Mori, 2021).

79-80 What does Tifaki Hazi do? What type of agency are they? I'd choose a different word other than 'badly' sensitised. I'm not sure who DIRMOM are, and it's not explained or referenced.

→ The numbers we refer to come from an internal report given to us by one interviewee. Tifaki Hazi appears in that report as a local agency mandated by the state services. However, we could not find the original report from the agency, nor could we find the agency itself on the internet, so we removed the reference to that source from the new version.

92 Who are the 'official sources'? This needs a bit of clarity.

→ We removed that part of the sentence while reworking the introduction.

93 Very interesting about the STTM Facebook group (I'm going to go and read Laura's paper!), perhaps a footnote could be added to give a little more information about this group?

→ We added more explanation about this group in the new section 3.

104 I don't see how the newspaper headline highlights an inability of experts to document the felt earthquakes. I'm making the assumption that the experts were not unable, they merely may not have communicated effectively, or at all. If so, could you make this clearer?

→ We removed that part of the sentence while reworking the introduction.

112-114 This whole quote is interesting. Is the author of the quote suggesting that the population may be evacuated ('leave the island')? Has that really happened in the past, as the quote seems to suggest? Obviously for reasons other than seismic activity! [This is merely a point of interest for me and nothing for the authors to change in the manuscript]

→ We tried to explain better in the new version of the introduction. We added a footnote explaining that, in the quote, "we" refers here to the civil servants coming from metropolitan France to work in the overseas department of Mayotte.

118 Please could you add the estimated volume of lava erupted? I would also suggest clarifying this sentence in line with the results from Cesca et al – it was the largest geophysically monitored submarine eruption to date. The comparisons with the subaerial eruption of Laki make for confusing reading.

→ We rephrased the sentence. We removed the reference to Laki and added the most recently published estimations of volume for the new underwater volcanic edifice (from Feuillet et al. 2021).

123 The activity itself was not new, but in May 2019 the population learned what the seismic activity was related to. Please could you clarify this.

→ We rewrote this part to avoid confusion.

133-156 These paragraphs seem out of sync with the previous. If you're reporting on events chronologically, this should come earlier, or perhaps consider putting this as a separate subsection?

→ We rewrote the whole section. These paragraphs have mostly disappeared. The remaining bits and pieces are now in the discussion where we think they integrate better (new section 7).

159 Please revise your use of risk reduction cycle. There is no cycle of risk reduction – you may be confusing it with disaster risk management? We should really not be thinking of cycles at all in this regard (maybe use 'phases' instead?), as any disaster management efforts should be improving what has come before.

→ Perfectly agreed. It was a mistake due to bad translation. The sentence was removed in the new version.

170-174 The sentences about scientific method determining public decision making needs some further work. These studies to which you refer are not suggesting the scientific method itself determines/supports public decision making about risk as much as the wider socio-political context. It's the evidence itself, the uncertainty and how it's presented, and possibly also the scientists involved….but not so much the method…? I agree it is very complex though! As a follow on, please could you revise your suggestion of the motivations of 'social volcanology' - which aims to sensitise researchers to the social context affected by or at risk from the volcanic hazard, and adopting methods commonly used in the social sciences to try and understand the social context better. I may have misunderstood what you are trying to discuss here, but I would urge you to be clearer. I would encourage you to look into work by Crowley, Hudson-Doyle, Haynes, Bird, and Hicks (all separate studies) to broaden and enrich this discussion (here and later in the paper) of the wider literature in volcanic risk communication. It is too brief at present.

→ We removed that paragraph. We added more references (and more explanation) in the introduction and in the discussion. See our response above, in page 2, section 1. We hope the reviewer will find the modified version clearer.

192-200 I agree it would have been interesting to find out if the information communicated by the scientists and authorities helped people to adapt their response to the crisis, but no data has been collected to that end. The study reports on a 'feeling' of improved communication by scientists from 2019, but not behavioural changes in the population as a result of the improved communication. I'd advise you to be careful about what they are, and are not, presenting evidence for, and as such perhaps revise parts of this paragraph.

→ The paragraph was rewritten in the new version. We think things are clearer now. The formulation was, indeed, misleading.

201 The figure could be improved a little to match the superb standard of the other figures (e.g. no need to use bullet points in a title and font sizes on the axes could be revised to draw attention more to the data and data captions itself)

→ We removed that figure from the new version. After reflection, we thought it was not really useful to make our point. We moved the point in itself in the result section (section 6) where it fits better.

213 I strongly urge you to think carefully about you use of social sciences here. Is this truly a social scientific study?

→ See our response above, in sections 1 and 2. We hope the new version of the introduction and methodology sections answer by themselves to this remark.

216 What does factually analyse mean? This is not an analytical methodology used in the social sciences.

→ Good point. The sentence was removed as we reworked the methodology section.

256 Incorrect use of the word (and approach) coding. Coded data refers to a specific type of qualitative data analysis to find common themes and concepts. You have simply ordered the data by data of publication and publishing author, and not extracted themes.
+
277 I don't see how you have quantitatively analysed anything here. You have visually presented the quantitative data using R, but you have not used the package to analyse data. The visual presentation of data does help you to make inferences and suggestions about how timing of earthquakes is related to communications from the scientists and authorities, but there are no stats, no modelling, no quantitative analysis. I would suggest that you don't inflate your methodological approach! This is, in my view, a completely qualitative study – it might be mixed method – but it's still qualitative….and that's ok!

→ Agreed. The sentence was removed as we reworked the methodology section. As explained above (see page 3, section 2), we did make first order thematic coding to analyze the content of the documents we collected and the evolution of this content. But, we did not use quantitative discourse analysis methods. This is something that we are working on now and it is out of the scope of the present research paper. That is why we did not want to expand too much in the previous version. The explanation is probably more balanced in the new version.

292 – 294 For me it is less interesting to find out the length of the interviews as it is how representative these interviews are of the institutions that the individuals represent. How many people work for these institutions?

→ Agreed. We tried to be a bit more specific in our revised manuscript but we needed to maintain interviewee's anonymity. This is something very difficult as the community we are working on is very small. It is very easy to identify people if we give too many details. So, we settled for presenting their functions.

295 Thank you for including information about the types of questions you asked in your interviews. This is really useful!
→ Thanks

312 What ethical procedures did you have in place prior to conducting the interviews? Did you acquire written or verbal consent from interviewees? Ethics assessments are crucial when using human subjects for data collection.
→ Most interviews were conducted via visioconference because of the restrictions due to the COVID-19 pandemy. All interviews were recorded with the verbal agreement of the interviewees and transcribed soon after. The transcriptions were anonymized when used for discussion between the members of the team (only the first author has access to the original files as she was the one conducting the interviews). Citations taken from interviews for illustration in the present paper are anonymized to respect interviewees' confidentiality. We also provide our own English translation. These ethical procedures are now explained in the revised methodology section.

587 You state that the anecdotes have been 'quite commented on' within the scientific community. Where? How many comments? Who made them?
→ This part was rephrased and partly moved to the discussion section (now section 7) so the comment does hold anymore. But, to answer briefly to the reviewer, our initial statement comes from participant observations. It is not easily quantifiable but it was reported by several interviewees. We could have references to these interviews.

588-591 You mention earlier in this paragraph that the communication is marked by a sense of surprise, but the quotes you use in this paragraph do not demonstrate surprise. Another quote or two would be useful here to support your narrative.
→ This part was rephrased when reworking the result and discussion sections (now sections 6 and 7) so the comment does really hold anymore. The objective here was really to insist on the second and third part of the sentence i.e. "uncertainty and even sometimes inaccuracy" rather than on the "sense of surprise" (already discussed earlier in the previous version).

639 The sentence 'important role in raising awareness of the importance of' leaves me hanging – what and why was it important? Did the scientists really not think this activity was important prior to the publication of Briole's blog?
→ Scientists interested in these phenomena certainly did, but - as we all know - institutions are slow to move. Many believe that these notes, and the November 11 event that brought international attention to Mayotte's event, contributed to creating the momentum that allowed to accelerate collective motion.

674-678 You say this is worth pointing out, but I'm not sure why? A bit more information is needed here.

→ We modified the sentence. We meant: it is worth pointing out that the volcanic hypothesis is now put forward in the official communication.

690 reinforce monitoring and prevention measures? What type of prevention measures? Risk reduction? Mitigation? Please be specific.
→ We added the following footnote: "The press release indicates that the government has defined the following action plan: 1) Complete as soon as possible the monitoring system and install the scientific devices that are necessary to continuously monitor the phenomenon; 2) Complete, through appropriate missions, the scientific knowledge; 3) Immediately update the knowledge of the risks presented by this phenomenon and the potential impacts for the territory of Mayotte; 4) Strengthen without delay the planning and preparation for crisis management ; 5) Regularly inform the population, in conjunction with local elected officials."

704 Communication of what to whom? Please be specific.
→ We meant public information. We modified the sentence accordingly.

713 I'm not sure what you mean by "it corresponds to the bulletin of July 2019". Please could you clarify this?
→ Several bulletins are issued approximately at the same time (one for July and two for August 2019). We modified the sentence to make it clearer.

734 Could you add some quotes here from your interviews to support your understanding of the tensions that existed between scientists and authorities?
→ We could not find a usable quote (explanations are often quite lengthy and not very easy to insert within the text) but we added a reference to the interview giving more detail about this anecdote and a sentence giving the output of that internal "crisis". The poster was actually withdrawn.

821 Live comments on social media are interesting – have you a sense of whether the sceptical comments are made by a vociferous minority or representative of the wider population?
→ It is difficult to know.

856 We may remain at odds here, but I would hesitate to use analysis in the truest sense. Had you coded the dataset into a set of themes and explored them (i.e. as in a thematic analysis), then perhaps. Perhaps you could argue it is a narrative analysis, as in you have presented a narrative of chronological events with some quotes from interviews, but I don't see a robust social scientific analytical methodology. Also, I'm not sure what you mean by quasi exhaustive documentation? Do you mean you looked all available communication about the events, written, verbal or otherwise? It would be great to see examples of the different types of documentation you looked at.
→ We hope that the new introduction and methodological sections answer those concerns. We adapted the first sentences of this paragraph to be clearer on what we mean by "analysis" (now section 7).

862 What is the evidence for persistent discontent of the population?
→ The sentence was rephrased while reworking the first paragraph of that section. We now refer to the perception of a lack of information and refer to the work of Fallou et al. (2020) and to the new sections 3 and 6 where this perception is discussed in more detail.

877 Rephrase 'information that adapts to the level of perceived danger'. It is very unclear what you mean here. Surely any perception of danger comes from personal knowledge/experience?
→ The sentence was removed while reworking on the discussion section. See next comment.

911-913 I can't see how this paragraph leads into subsequent sections on bias? I would suggest guiding the reader through these sections a little better. Perhaps the word bias is the confusing factor…
→ We reworked the introduction of the discussion section to develop more on what is known in the litterature about at-risk populations' information needs (new section 7). We conclude saying that "the public information strategy that has developed over time has not avoided some well-known pitfalls of risk communication that would benefit from being corrected in the future." We believe that it better introduces the idea of bias.

958 What evidence do you have that the 'population's capacity for resilience increased' as a result of sharing experiences on Facebook?
→ We removed that sentence from the new version of the manuscript. The formulation was misleading.

975-980 I'd highly recommend taking a look at the work of Hudson-Doyle here.
→ We did and we added a few sentences about some of her work (very interesting indeed!).

1107 Please clarify what you mean by a gap in culture between scientists, authorities and society. Motivations? Concerns? Needs? Knowledge? Worldview?
→ We rephrased that sentence to gain clarity. We also added references to earlier works.

1015-1017 Risk scenarios have been mentioned at times throughout the paper, but not really explained. In this sentence are you suggesting that scientists have been focused mainly on the hazards and not so much on the risk? Are they actively working on risk assessments (i.e. assessing vulnerability, exposure etc), as you seem to suggest? Are they quantitative/qualitative or both? Who are they collaborating with on this? This is interesting!
→ The work on the scenarios is still in progress and the list of possible scenarios has not yet been published by the REVOSIMA so we cannot say much more in the paper. The latter is really based on the information that was made public.

1040-1042 Here's those three papers again….I'd advise demonstrating that you've read into the subject of volcanic risk communication more extensively.
→ Those papers are actually quite relevant to what we wanted to say. We tried to add a bit more diversity with another reference. We hope that the changes made in the introduction and discussion sections will answer the reviewer concern.

**Referee 1: Anna Hicks' Technical corrections**
Please check all references – there are many typographical errors.
Consistency throughout with use of the word 'seismic' – sometimes you use sismic/sismo
There are several typos and grammatical corrections required. I've outlined a few below, but recommend a detailed proof read prior to the submission of the next version.
+
2 I would suggest using a different word other than limits. The paper does not really talk of the limits of risk communication in this case. Lessons, perhaps?

17 was shown
166 Please add a reference to this sentence
169 Add making to decision
248 remove 'made up'
559 Experts appeared puzzled to whom? And were they genuinely puzzled, rather than apparently?
564 publishes daily reports via where? Their website? Would be good to add a link here.
599 inaccurate rather than inexact
601 is at odds
616 remains difficult is a bit ambiguous – I'd suggest a change of phrasing here.
624 Amongst the [incorrect] (or use other word) explanations…
637 November and December.
638 add (sic) after "wild"
722 I would hope that the national media were not stoned! Please could you adopt a different phrase?
914 Bias misspelling
1113 reach not cross
1123 acting qua?

→ We implemented all the requested changes that were left after we had answered the previous comments.

**Referee 2: Julie Morin' specific comments**

l.18 – People are exposed to hazards, not to risks
→ The abstract was completely rewritten. This comment does not apply anymore.

l.19 – are local/national authorities – or even the scientists - included in "people" if the absence of known historical event is the criteria to suggest naivety?
→ The abstract was completely rewritten. This comment does not apply anymore.

l.31 – I would remove "very" which is questionable (very active in regard/compared to what?)
→ Agreed, but the introduction was almost entirely rewritten and this comment does not apply anymore.

l.34 – detail what BRGM means. Please be explicit and consistent each time you refer to a new acronym in the paper (the whole name, always either in English or French – pick up one of the languages and stick to it)
→ Agreed. We made the corresponding changes.

l.42 – new insights to be found in the papers recently published on Mayotte e.g. Dofal et al. 2021?
→ We now summarize the results of many new papers (a lot of them published after we submitted) in the new section about Mayotte's geological setting and ongoing activity (section 2 of our revised manuscript). These papers - 9 published in 2020-21, including the one by Dofal et al. - are now cited together with some older ones about the geological context.

l.43 – an active volcano monitored by a network probably rather than studied
→ Agreed. We made the corresponding change.

l.45 – you could probably mention that Karthala erupted several times in the past decades (see Bachelery et al. 2016) as it has been providing a regional experience of volcano crisis management and communication

+

l.77 and 78 – is it necesseraly true that there is no human memory given that 45% are Comorian and that the Grande Comore Island counts an active volcano with rich historical activity?

→ Agreed. We added a sentence in the new section 3 specifying that people coming from the neighboring Comoros islands may have experienced previous seismic and volcanic crises as four eruptions occurred in 2005, 2006 and 2007 (Morin *et al.*, 2016).

l.63 – hazards instead of risks

→ The sentence was removed in the new version of the introduction so the comment does not hold anymore.

l.69 – since when are the episodes of civil unrest happening?

→ We reworked that paragraph and added more details about the social context and the episodes of civil unrest. Since 2011 (date of the departmentalization), Mayotte has been regularly shaken by social crises. The most recent one, which brought the economy to a standstill for two months in the spring of 2018, was just ending as the first earthquakes began (Roinsard, 2019; Mori, 2021).

l.69 – Can you provide a reference to sustain the population's distrust statement?

→ Same answer as for Anna Hicks' comment. We reworked the corresponding paragraph to add a bit more information about the social context of Mayotte. Distrust in state services is quite common in many of the French overseas departments, relationships with metropolitan France being complicated by the colonial past. This is something that is often briefly mentioned by authors working in these areas. Mori (2021) mentions it in her recent paper about the seismic crisis in Mayotte. It is also mentioned by Komorowski et al. (2018) in "Challenges of Volcanic Crises on Small Islands States". In Mayotte, the context is however a bit different from the other French overseas (see Regnault, 2011). Anthropologists agree that there is no real integration between the traditional culture of the villages and the more westernized culture of large cities (Lambek, 2018). According to Regnault (2011), "three quarters of the *Mahorais* - rural or, at least, still very attached to their village - live a culture other than the "westernized" culture of the cities" (*trad. by the authors*). The relationship with state authorities is complicated by the island's colonial past, but also by a sense of disappointment among the population, who expected more rapid changes to bring the island up to French standards after departmentalization (Roinsard, 2019). As stated earlier, since 2011, Mayotte has been regularly shaken by social crises. The most recent one, which brought the economy to a standstill for two months in the spring of 2018, was just ending as the first earthquakes began (Roinsard, 2019; Mori, 2021).

l.79 – what is Tifaki Hazi? An agency specialized in opinion surveys? To be clarified

→ Same answer as for Anna Hicks' comment. The numbers we refer to come from an internal report given to us by one interviewee. Tifaki Hazi appears in that report as a local agency mandated by the state services. However, we could not find the original report from the agency, nor could we find the agency itself on the internet, so we removed the reference to that source from the new version.

l.80 – explicit what DIRMOM is

→ Agreed and done.

l.93 – explicit here that it is a seismology citizen group as you do l.136

→ We added more explanation about this group in the new section 3. We do not use the term "seismology citizen group" because it refers to discussions that are out of the scope of the present study.

l.94 – "quickly become very active in the public discussion": what does it mean? What is the frametime? How many posts? How many people involved + which proportion of the population are represented?

→ We added more explanation about this group in the new section 3. We believe it make things clearer.

l.119 – Can you provide an average value?

→ Same answer as for Anna Hicks' comment. We added "more than 5 km3" citing Feuillet et al. 2021 which is the most recently published estimations of volume for the new underwater volcanic edifice.

l.129 to 131 – Why are your stating that this announcement only partially meets expectations?: how? Which specific expectations are met or not? What does change with and after this announcement? Please provide the full meaning of REVOSIMA in French.

→ That sentence was removed from the new version of the text. We reworked the paragraph to illustrate better the feeling of a lack of information, before, but also after, the creation of REVOSIMA. This is also discussed in the discussion section (new section 7).

l.152 to 155 – Refer to the section of the paper in which you detail this (at this point, as a reader I am just thinking "but is communicating/supposed to communicate/ and how?")

→ Agreed. We removed that part when we reworked the introduction. The new section 3 settles for making the point of a lack of information and does not expand on what is exchanged within the STTM's facebook group. It is now discussed in the discussion section (new section 7).

l.170 – it is now widely recognized that there is no such thing as "natural risks", natural should be put in quotes

→ The sentence was removed in the new version of the introduction but we agree. And we did write "natural" risks in the conclusion.

l.177 – There is a need to develop further the references here. Other papers specifically focused on volcanic risk communication can be included.

→ Agreed and done while rewriting the introduction and reworking the discussion section (new section 7). See our response above, section 1, page 2.

l.183 – From part of the population: can you specify if this is either/both a significant number and a representative sample of people?

→ The sentence was removed in the new version of the introduction.

l.202, Figure 2, "Feelings of the population" section – Did Fallou et al (2020) quantify the feeling of improved communication, and how? Can you relate their results and/or explain the scale you have adopted?

→ The sentence and the figure were removed in the new version of the introduction. Fallou et al. (2020) did not "quantify" the feeling of an improved communication but they reported on the basis of their work on the social network. We removed the figure from the new version. The use of a mixed scale (quali/quanti) was indeed misleading and, after reflection, it was not really useful to make our point.

l.213-221 – I think the paper would greatly beneficiate from developing further the literature review on several topics: - volcanic hazard and risk communication is widely studied in fundamental papers by (e.g. but not limited to): Barclay, Bird, Bretton, Gomez-Zapata, Haynes, Hicks, Hudson-Doyle, Thompson, etc. - risk communication on "invisible" threats - lessons learned in terms of volcanic risk communication in other small island/oversea territories contexts

→ Agreed and done while rewriting the introduction and reworking on the discussion (new section 7). See our response above, section 1, page 2.

l.252-255 – It is unclear at this stage (and need to be clarified) if these 4 stages correspond to qualitative or quantitative gaps / evolutions in the response/decisions/policies. It is specifically confusing to describe the communication aspects as phase 4 while risk and crisis communication are necessarily involved at all stages (should they be considered as effective or not).

→ Agreed and done while rewriting the introduction and the methodology sections. See our response above, section 3, page 3.

L.276 – Do you mean "ordered" or "classified" rather than "coded"? Unless I misunderstood the methodology, I think the data was not coded. If it was, there is a need for describing this further in detail.
+
l.281-283 – what type of correlation did you do, which main specific functions of R did you use?
+
l.287 – Specify how you did make the content analysis (which type of coding have you used, etc.)

→ The methodology section was almost entirely rewritten to account for both reviewers' comments. See our response above, section 2, page 3.

l.291 – Please provide a grid (table of bullet points) of the semi-structure interviews, with at least the main questions

→ We added more details about the questions asked during interviews in the new methodology section.

l.294-295 – Please harmonize the way you refer to the actors. The Ministries names are provided in French l.85-86 while here they are in English, which might lose your reader.
Same l.457 to 459. + l.686

→ Well spotted. We made the corresponding changes.

l.303-305 – I think it is important to mention if those selected citations are representative/randomly selected/picked-up specifically to illustrate your statements, etc., that is to say to further explain how you discriminated the citations you chose to use (e.g. did you focus only on the descriptions of the

communication gaps, or did you make a full qualitative content analysis and then picked-up a fe topics – in which case it has to be described).

→ Agreed. As stated in the text, we explored STTM's Facebook publication feed but without aiming for exhaustiveness as it was difficult to achieve without adequate tools. We use excerpts from STTM facebook posts to illustrate some of our statements and do not aim to provide a full qualitative content analysis. Fallou et al. (2020) provide a more extensive description than us as their paper is really focussed on that facebook feed.

l.307-311 – These details should be provided much earlier in the paper, when you first mention STTM (and you could just write here something like "As stated before, STTM is not representative of the whole population…")

→ Agreed and done. We added more in the new section 3.

l.313 - An additional figure showing the organisation of governance and the theoretical channels of communication between the different actors (including citizens) would greatly help the reader going through the paper (maybe in two times: before/after REVOSIMA was born?). It should be placed early enough in the paper to go easily through all the acronyms provided.

→ Very good idea. We will submit the corresponding figure with the revised manuscript (it will be called Figure 2 in the new version of the text).

l.322-323 – Please explain better why you did chose to ignore the Mayors' communication: was it too complicated to collect the few communication they did? Or did you collect it but the low number or the contents are too anecdotic to allow this data being analysed? It is important to explain this as many other actors have not communicated much during the crisis as evidenced in Table 1.

→ We agree with the reviewer. In the French system, mayors are usually key actors of risk and crisis management. But, in the case of the seismo-volcanic crisis of Mayotte, it soon appeared that public information was mainly being orchestrated at the departemental and national levels. The explanation that was given to us by interviewees is that the initial crisis overwhelmed the capacity of response of local mayors requiring the intervention of the prefecture of Mayotte, with the support of the national level. When analyzing the coverage of the crisis by the local press (Devès et al., under review), Mayors are only mentioned in the very beginning of the crisis when schools had to be closed because of the appearance of cracks in vulnerable buildings. Therefore, we decided to focus on the communication of the state services, which is by far the largest in volume. We know that some public conferences were organized by the prefecture with the support of the municipalities, but there are no written traces to be studied.

l.363-366 – Is the decrease in the number of felt earthquakes real or could it be that the citizen got tired of reporting it or other events on the island make the reporting process less likely? You could probably add a sentence to make this clear.

→ This is developed in the new section 2. There was a sharp decrease in the number of felt earthquakes after June 2018. The observed decrease is in line with the decrease in the number of instrumentally recorded earthquakes and of their average magnitude (e.g. Lemoine et al. 2020, Bertil et al. 2021). EMSC-CSEM catalog reports only ~4 felt events per month until the end of 2018, and then a moderate recovery in the number of felt events between February and June 2019 (~9 felt events per month on the average).

l.382 – Specify if the territory to which the lock down applies (island/national scales?)

→ The lockdown periods that are shown are those of metropolitan France (note that most of the scientific institutions involved in monitoring are located in metropolitan France). Mayotte endured longer lockdowns in spring 2020 and 2021 but there was no proper lock down in autumn 2020.

l.390 – Local/national authorities?

→ We rewrote the sentence to make it clearer. It now reads as: "at the beginning of the crisis, BRGM Mayotte is the natural interlocutor of the local and national authorities for decision support."

l.393 – What do you mean by "voluntary commitment"? BRGM and other institutions have no legal obligation to provide interpretations to the authorities? Please specify.

→ Yes, BRGM and some other institutions (it is the case for observatories) have a legal obligation to provide interpretations to the authorities, but mandates are not always very clear about the extent of these obligations. Nothing is said, for instance, about night and week-end shifts. However, this discussion is out of the scope of our study so we removed the sentence.

l.439-440 – Detail what OBS and GNSS mean and which sort of data they are intended to provide.

→ For OBS, this is now done in the new section 2. We do not repeat it later. OBS = Ocean Bottom Seismometer. We now develop the acronym for GNSS = Global Navigation Satellite System.

l.460-461 – If you do not detail this later in the paper, please provide the information here: how many schools/classes/kids involved? Who won the competition? If the name has not been displayed yet, you could maybe provide a footnote displaying the reasons why.

→ Agreed. We added a footnote that reads as follows: "The name chosen for the new volcanic edifice was finally made public in December 2021. It did not match the names originally proposed by the children. It is not possible to explain the reasons for this in this paper, as it would require extending our study period. However, it can be noted that the entire process was not consistent with the need to engage people more actively in the recognition of this new source of hazard." We plan to address that issue in a specific paper.

l.507 – After reading the whole section, I would suggest to replace "analysis" by "description of the scientific and official communication" (same l.526).

→ Agreed. We changed the name of the section while reworking the introduction. It is now: "The organisation of the process of public information and its evolution", which implicitly corresponds to a description of "the organisation of the process…".

l.529-531 – I think it is important here to better explain how you have defined the 3 phases ABC (qualitatively, quantitatively, which criteria?). For instance, currently, when looking at Fig 4, I immediately wonder why the phase A does not stop when the very intense phase of communication by only two main actors ends in July 2018.

→ We added more explanation. See our response above, in section 3, page 3. The definition of the phases A, B, C corresponds to changes in the modalities and/or content of public

information and not just to changes in the volume or frequency of the communications. The change you notice as an example here does not correspond to a significant change in the modalities or content of the communications. It is simply associated with the decrease of the number of felt earthquakes. There are consequently less press releases but their content does not change significantly.

l.553 – Communication being one major component of thr risk management process, you might want to add this first public press release on Fig 3.?
→ Agreed and done.

l.564 – Is the switch to "crisis monitoring" only a communication process or a whole change of organisation to deal with an ongoing process (and in that case what does this imply?)
→ In this case, "crisis monitoring" implies analysing the data as quickly as possible and providing the authorities with information on a daily basis, and for every major event (even during the week-end).

l.575 – "expert body" = do you mean BRGM, name clearly the experts involved to avoid any confusion
→ Agreed and done.

l.611 – I am not sure that the example provided (falling down stairs) can be qualified of nor associated with the idea of misbehaviour (the current research on disasters produced by social psychologists develop the idea that any behaviour is sustained by personal logics themselves rooted by many factors and that there is generally no such thing as a misbehaviour – even when the behaviours do no align with the ones the authorities or other people would except). For that reason, I would probably just say behaviour.
→ We agree. We took the term from the experts' report and should have put it between quotes. We reformulated the sentence (because the report is not public).

l.706 – Could one example of leaflet be provided in the supplementary materials? If you know about it, you could probably add a sentence on the way this new documentation has been perceived by the citizen.
→ Yes, we can provide a few examples in supplementary material but we do not know how these were perceived, nor if they were seen by much of the population.

l.723 – Is directly mentioning a name aligned with your ethics approval?
→ This person is mentioned explicitly in the press article that we refer to, but we removed the name from our text.

l.728 – Municipalities instead of communes?
→ Yes indeed. We modified it.

l.739 – What do you mean by "But the case is quickly closed"? Please, clarify.
→ We removed that sentence while reworking the text, but we added a reference to the corresponding interview and a sentence giving more insights on the output of that internal communication "crisis".

l.750 – Can you provide a number of occurrences to allow the reader understanding what "largely commented" means?
→ No we cannot because undertaking an exhaustive analysis of facebook feed is impossible without specific tools. Plus, it was out of the scope of this paper. We removed 'largely'.

764 – Probably specify released "online" because the reader is wondering how it is released until having the information at the end of this paragraph
→ Agreed. We modified it.

l.815 – what are these possible scenarios?
→ Same answer as for Anna Hicks' comment. The work on the scenarios is still in progress and the list of possible scenarios has not yet been published by the REVOSIMA so we cannot say much more in the paper. The latter is really based on the information that was made public.

l.850-852 – Do you know if the STTM network had an indirect access to it, and/or how the documentary is perceived by the scientific, authorities, and citizens?
→ We did not specifically focus our interviews on that event but it is an interesting point. According to the journalists we interviewed, the coverage of the event was made very hard by the absence of visuals of the activity. One of the main contributions of that webdoc was hence to provide a whole set of new visuals that are still used today by journalists and scientists to communicate. We added a few sentences about that in the text.

l.879-888 – How are these points developed by Lindell and Mileti concretely translated in Mayotte? It currently looks a bit disconnected from your case study. To better "follow" your reasoning, it would be very helpful to read a summary of the different questions addressed (or not/ or partially) for the Mayotte case during the different phases you have identified. This summary might eventually include, as you have done it in other sections, the objective communication provided by the actors vs. the perceived communication by the population.
→ We reworked the introduction of the discussion section to develop more on what is known in the litterature about at-risk populations' information needs (new section 7) and make the link with Mayotte's case study clearer. We also added a few more references.

l.894-896 – Is this thirst confirmed through the STTM content analysis? Maybe remind if you mean a qualitative, quantitative thirst, or both.
→ As explained earlier, we explored STTM's Facebook publication feed but without aiming for exhaustiveness as it was difficult to achieve without adequate tools. We use excerpts from STTM facebook posts to illustrate some of our statements and do not aim to provide a full qualitative or quantitative content analysis. Fallou et al. (2020) provide a more extensive description than us as their paper is really focussed on that facebook feed. In this paper, we settle for comparing what we learnt about the public information process with what is known about at-risk populations' information needs in general, illustrating typical difficulties by excerpts taken from STTM's Facebook feed.

l.931-932 – Is the increased difficulty to understand data uncertainties your own interpretation or is or based on a content analysis? Please make it clear.

→ We reworked that sentence to make it clearer that it was a comment on what precedes. It now reads: "Although this strategy seems legitimate from a scientific point of view, one can wonder if it really helped people to better understand the nature of the existing uncertainties."

l.951-954 – I think this side note on mass panic is a bit out of the scope as it apparently does not apply to Mayotte territory, where the problem is "just" a worried population seeking for information. I would probably remove this.

→ We reworked that paragraph and removed the sentence.

l.958 – The capacity for resilience has to be proved. I would probably just talk about coping here, more than resilience.

→ Same answer as to Anna Hick's comment. We agree and we removed the sentence.

l.999 – risks = hazards ??

→ Nope. We mean risk and it is a verbatim of one of our interviews.

l.1007 – please provide a short summary of those scenarios (and if this is meaningful, when each scenario was established/evolved)

→ Same answer as to Anna Hick's comment. The work on the scenarios is still in progress and the list of possible scenarios has not yet been published by the REVOSIMA so we cannot say much more in the paper. The latter is really based on the information that was made public.

l.1044 – "they have a moral, if not legal": please provide a footnote to explain the sismovolcanic science/legacy aspects in the French territories (till which end are the scientists/authorities legally responsible in terms of information?)

→ Well, that is a good comment indeed, but it is out of the scope of this paper to discuss these issues. It would add a lengthy footnote and the reader would not learn much more than there are institutions or observatories running under convention that have a well-defined mandate with respect to crisis decision support and others who run without clear convention concerning that issue. The question of their implicit mandate arises then, but cannot be answered easily.

l.1079-1080 –I think I disagree with you on this specific point. How do you discriminate the part of the population able to understand? Isn't it the role of good scientific communication to provide several levels of reading so that everyone has access to the information?

→ We removed that part of the sentence. It was not important to make our point and open a whole new discussion we don't want to have at this stage of the paper.

l.1093-1100 – It would greatly strengthen this paragraph to provide some bibliographic references, as the literature on disaster risk reduction includes many common arguments.

→ We reworked the paragraph and added a few references (in addition to the ones added in the introduction). Do not hesitate to propose more references if you still find it necessary.

**Referee 2: Julie Morin's Technical corrections**

l.2 – should be either "during sismo-volcanic cris**es**" or "during **a** sismo-volcanic crisis"

l.381 (figure) – in phase 3 : ga**s**, and phase 4: gouver**nm**ent, det**ec**tion, ga**z**

l.388 and 404 – 3 = Three You mention l.439 OBS while l.450 OBS**s**

l.441 – University of **La** Reunion

l.488 – 8 = eight

l.500 = lock down**s**

l.637 – **N**ovember **D**ecember

l.689 – 4 = four

l.859 – a**n** inter

→ We implemented all the requested changes that were left after we had answered the previous comments.

---

## Author Response (AR2)

Authors' answer to Reviewer's report

Congratulations on writing a very interesting and important publication and thank you very much for kindly taking each of my considered notes on board and providing a detailed response. I'm glad to hear that the reviews were helpful. The manuscript has certainly improved considerably and I am delighted to support its publication asap, and pretty much as is. To speed up the copy editing process, I have provided a list of very small amendments (typos mainly) below, which should take you an hour at most to complete.

We would like to thank Anna Hicks for her very positive evaluation of our manuscript, and for her detailed comments during the full review process that greatly helped us to improve our paper's quality and soundness.

We list below our answers to Hicks' last comments, and we append a marked changes manuscript at the end (changes indicated in red).

Looking forward to seeing it in print and referencing it in my work!

62: What do you see as the difference between human and social sciences? Perhaps drop human?

Done and throughout: It's very rare that sentences can start with 'Because'. It's seen as poor grammar in written English. Perhaps consider going through the manuscript, and adopting other synonyms, such as 'as' or 'however'.

Done. Changed to "As" on line 122. It was the only sentence starting with "Because" in our text.

170-176: I'm no seismologist, but it is a little confusing to have ML and then Mb. I'm assuming you used Mb due to the regional seismicity, but perhaps something you should clarify.

The events close to Mayotte that happened since the start of the seismic sequence in May 2018 are indeed reported with $M_L$ in Lemoine et al. and REVOSIMA catalog (plus $M_W$ for the few largest events). For the regional seismicity (blue dots on the map of Figure 1), we used the NEIC-USGS catalog. We checked again this catalog to find that it reports magnitudes either as Mb or as Mw, depending on the events (plus two old ones reported with Ms). We thus modified our text to "…with the largest magnitudes recorded between 5 and 5.5 ($M_b$ or $M_W$ according to USGS catalog)."

183: change to 'allowed the location of'

Done

229: change to 45% are from the Comoros

Done

343: sensibilisation comes from French, I would opt for sensitization or awareness-raising

Changed to "awareness-raising" as suggested

349: video not vision

"visioconference" changed to "videoconference" as suggested

350: pandemic

Done

Fig 2: Interesting that there's no overlap between the groups. Just something to think about!

Fair remark. Nothing to change in our text or figure, but yes, we agree that it's something to think about in the future.

375: there's a rogue italic p corrected and throughout: Similar to 'because', it's rare we start sentences with 'But' in written English. It's ok now and again, particularly when the style of writing is more narrative, but the 'but's' increase in number throughout the manuscript and it gets a little too much. I would go recommend going through and adopting other words now and again.

We changed this occurrence to "however". We checked for similar issues through the text and modified more than half of the occurrences of "but" at the start of sentences.

and throughout: Please go through the manuscript and check all your use of numbers against words. All numbers under 10 should be written as words, except of course for dates, magnitudes etc. Additionally if you begin a sentence with a number, even one over 10, it must be written in full.

Done

549: as an example of above

OK. Done

565: delete the extra space between scientists and also

Done

593: perhaps provide a footnote for what a pickathon is?

We added a short footnote as suggested

600: capital December done

601: be consistent with your use of euros or Euros done

604: what do you mean by actors here. That's the first use and it feels a bit odd

Changed to "scientific monitoring actors" to clarify.

621: you say average number and then provide a range. Maybe say was between 6-8 etc

We infer the reviewer misunderstood us because the decimal values were written with a comma (as in French) instead of a decimal point. Sorry. We fixed this issue.

668: be consistent with capitalisation (or not) of prefecture fixed

669: space between published and press done

686: remove space before ; done

768: and the BBC

changed

794: needs an extra ) at the end fixed

812: you don't find out until the end of the para that it was a poster-based communication. I would suggest adding that before communication on this line.

done

853: quote in italics done

854: full stop needed after reference done

860: no 's' needed on pandemic corrected

894-895: interesting. Why do you think that was? Was it asked for by the public? Not needed in the paper, but I just find it fascinating. You mention a bit about scenarios later and perhaps this is something we could work on together for a conference session, maybe with UWI-SRC too?

Yes. Thanks for the suggestion. Nothing to change in our text.

923: Is this still the case?

No. It has been published in August 2021. We changed the sentence accordingly.

985-995: Strikethrough section remains fixed

998: Suggest changing to 'Before going further'

done

1037: This is the first time I've ever read the word complotism! I had to look it up! Nice!

No change needed

1106: Add Hurricane before Katrina done

1165-1167: I don't know what frilosity means.

Sorry, it's an anglicized French word. We changed it to "reluctance" in the interview's quote, then to "shyness" in the following sentence.

1228: capital March done

1281: trust not thrust fixed

1319: capital International.

done

WELL DONE! :-)

Thanks ;-)

[revised manuscript text omitted]